# The Transformer Cookbook

**Andy Yang**[*]                                                                              *ayang4@nd.edu*
*University of Notre Dame*

**Christopher Watson**                                                          *ccwatson@seas.upenn.edu*
*University of Pennsylvania*

**Anton Xue**                                                               *anton.xue@austin.utexas.edu*
*University of Texas at Austin*

**Satwik Bhattamishra**                                                    *satwik.bmishra@cs.ox.ac.uk*
*University of Oxford*

**Jose Llarena**[†]                                                             *jose.llarena@gmail.com*
*Independent Researcher*

**William Merrill**                                                                      *willm@allenai.org*
*Allen Institute for Artificial Intelligence*

**Emile Dos Santos Ferreira**                                                    *emileferreira@sun.ac.za*
*Stellenbosch University*

**Anej Svete**                                                                            *asvete@ethz.ch*
*ETH Zürich*

**David Chiang**                                                                         *dchiang@nd.edu*
*University of Notre Dame*

**Reviewed on OpenReview:** *https://openreview.net/forum?id=sPshCSvDrX*

## Abstract

We present the transformer cookbook: a collection of techniques for directly encoding algorithms into a transformer's parameters. This work addresses the steep learning curve of such endeavors, a problem exacerbated by a fragmented literature where key results are scattered across numerous papers. In particular, we synthesize this disparate body of findings into a curated set of recipes that demonstrate how to implement everything from basic arithmetic in feed-forward layers to complex data routing via self-attention. Our *mise en place* of formulations is for both newcomers seeking an accessible entry point and experts in need of a systematic reference. This unified presentation of transformer constructions provides a foundation for future work spanning theoretical research in computational complexity to empirical investigations in architecture design and interpretability. We provide code implementations of each construction in numpy alongside a suite of generative unit tests.

---

[*]Corresponding author. 🌐 Formal Languages and Neural Networks Seminars: `https://flannseminars.github.io/`
[†]Code maintainer. 🦎 Code `https://github.com/JoseLlarena/transformer-cookbook`

# 1 Introduction

Transformers are the key ingredient of large language models (Vaswani et al., 2017). While they are typically trained through data, there is also much interest in explicitly "programming" algorithms into their parameters. By linking algorithmic procedures to model constructions, we obtain a more transparent view of transformers: the problems they can solve, the mechanisms they might implement, and their fundamental limitations. In practice, theoretical results proven through transformer constructions are commonly used as motivation for tasks ranging from model training to mechanistic interpretability.

However, transformer constructions are difficult to study and apply. The literature is highly fragmented, with key results scattered across numerous publications, each with its own unique notation and architectural assumptions. For instance, a major step was made by Weiss et al. (2021) who created the RASP programming language, followed up by Lindner et al. (2023) who created a compiler that compiled a subset of RASP programs into transformer weights. However, RASP as a full language was somewhat more powerful than transformers, and compilation using Tracr required restrictions on the input length. While notable surveys, such as that of Strobl et al. (2024), chart and map out key results built from these constructions, they rarely dive deep into the specific technical details. In fact, there has been no unified presentation of the techniques used in these constructions, which forces newcomers to piece together the requisite knowledge from disparate sources.

To address this gap, we curate and systematize the common constructions of transformer-encoded algorithms, creating a unified reference for both aspiring and seasoned *transformists* that makes these methods explicit, precise, and accessible. While each construction comes with its own set of assumptions and preconditions, our unified presentation enables the reader to compare the detailed implications of each one. This synthesis establishes a common ground for both theory and practice: theoretically, it abstracts a set of core computational principles from the literature; empirically, it offers a library of idealized circuits to guide the design of new architectures and the analysis of existing ones. More broadly, we believe this unified perspective is essential for building safe and reliable AI systems.

To achieve these goals, this cookbook is structured as follows:

- **Preliminaries** (Section 2). This section gives an overview of the mathematical background and the key components of a transformer. Even if the reader is already familiar with this topic, we nevertheless encourage giving this section a read, since nearly all constructions in the book will follow the notational conventions introduced here.

- **Basic Ingredients** (Section 3). Before diving into constructions, we next present some lemmas that will be of use. Additionally, we discuss some key concepts that must be considered in any rigorous construction, such as the issue of uniformity and representing common data types.

- **Feedforward Layers** (Section 4). The feedforward layers of a transformer offer opportunities for rich computation. Here, we discuss how to construct arithmetic computations, logic circuits, and other tricks. A summary of the key constructions is shown in Table 1.

- **Self-Attention Layers** (Section 5). Innovations in the self-attention layer are what set the transformer apart from the architectures that came before it. We show how to construct useful aggregation and selection operations here. A summary of the constructions is provided in Table 2.

- **Layer Normalization** (Section 6). This is an important architectural component that enabled early practitioners to train large-scale transformers with numerical stability. We show how to leverage this component to perform operations, such as the amplification of signals.

- **Rounding** (Section 7). Whereas it is often convenient for humans to reason in discrete values (e.g., tokens), transformers fundamentally operate on continuous-valued inputs. We discuss various rounding tricks here that allow for easy conversion from the continuous to the discrete.

- **Assembly** (Section 8). Having introduced many constructions, we now give useful lemmas on how to connect them. The primary operations involve putting constructions in sequence ("serial composition") and in parallel ("parallel composition").

- **Examples** (Section 9). To round out our cookbook, we provide instructive examples on how classical constructions from the literature may be achieved with our recipes. In particular, we give examples with induction heads and the Dyck languages. A summary of these constructions is given in Table 3.

## 2 Preliminaries

### 2.1 Notation

We write $[n]$ for the set $\{1, \ldots, n\}$. For any true/false statement $b$, we let $\mathbb{I}[b] = 1$ if $b$ is true and $0$ if $b$ is false. If $X$ is a set, we write $X^*$ for the set of sequences over $X$, and $X^+$ for the set of non-empty sequences over $X$. Let $\Sigma$ be a finite alphabet. We write strings in $\Sigma^*$ as $w = w_1 \cdots w_n$. We write $\mathbf{0}$ for the zero vector or matrix, $\mathbf{I}$ for the identity matrix, and $\mathbf{1}$ for a vector or matrix whose entries are all 1. For any function $f \colon \mathbb{R} \to \mathbb{R}$ and any vector $\mathbf{x} = [x_i]_{i=1}^d \in \mathbb{R}^d$, we extend $f$ to $\mathbf{x}$ coordinate-wise:

$$f(\mathbf{x}) = f\left(\begin{bmatrix} x_1 \\ x_2 \\ \vdots \\ x_d \end{bmatrix}\right) = \begin{bmatrix} f(x_1) \\ f(x_2) \\ \vdots \\ f(x_d) \end{bmatrix}.$$

We write $f \colon X^+ \xrightarrow{\text{lp}} Y^+$ to state that $f$ is a function from $X^+$ to $Y^+$, and it is length-preserving, that is, it maps every sequence to a sequence of the same length. Slightly unusually (following Strobl et al. (2024)), we will often work with sequences of vectors in $(\mathbb{R}^d)^+$, which we write as $(\mathbf{x}_1, \ldots, \mathbf{x}_n)$. Two sequences of the same length can be added position-wise: $(\mathbf{x}_1, \ldots, \mathbf{x}_n) + (\mathbf{y}_1, \ldots, \mathbf{y}_n) = (\mathbf{x}_1 + \mathbf{y}_1, \ldots, \mathbf{x}_n + \mathbf{y}_n)$.

To avoid excessive subscripting and superscripting in complex networks, we adopt a "dot notation" which is nonstandard in mathematics but hopefully familiar from many programming languages. If $f$ is a function that depends on some parameters, we give the parameters names like $f.\mathsf{a}$ and $f.\mathsf{b}$. If $g$ is another function with parameters $g.\mathsf{a}$ and $g.\mathsf{b}$, there is no implied relationship between $f.\mathsf{a}$ and $g.\mathsf{a}$.

### 2.2 Transformers

A *transformer* is a neural network that, in this paper, defines a length-preserving mapping from strings to strings. An input layer maps a string to a sequence of vectors. Then the network processes sequences of vectors through multiple layers, each consisting of a self-attention sublayer followed by a feed-forward sublayer. Finally, an output layer maps the sequence of vectors to a string.

Here, we define the overall structure of the transformer as a function, before defining the individual components in subsequent sections.

**Definition 2.1.** A *transformer* is a function

$$\mathsf{tf} \colon \Sigma^+ \xrightarrow{\text{lp}} \mathcal{Y}^+$$

$$\mathsf{tf}(x_1 \cdots x_n) = y_1 \cdots y_n \quad \text{where} \tag{1}$$

$$\mathbf{z}_i^{(0)} = \mathsf{tf}.\mathsf{we}(x_i) + \mathsf{tf}.\mathsf{pe}_n(i) \qquad\qquad i \in [n] \tag{2}$$

$$(\mathbf{z}_1^{(1)}, \ldots, \mathbf{z}_n^{(1)}) = \mathsf{tf}.\mathsf{tl}^{(1)}(\mathbf{z}_1^{(0)}, \ldots, \mathbf{z}_n^{(0)}) \tag{3}$$

$$\vdots$$

$$(\mathbf{z}_1^{(L)}, \ldots, \mathbf{z}_n^{(L)}) = \mathsf{tf}.\mathsf{tl}^{(L)}(\mathbf{z}_1^{(L-1)}, \ldots, \mathbf{z}_n^{(L-1)}) \tag{4}$$

$$\mathbf{y}_i = \mathsf{tf}.\mathsf{out}(\mathbf{z}_i^{(L)}) \qquad\qquad i \in [n] \tag{5}$$

where $\mathsf{tf}.\mathsf{we}$ and $\mathsf{tf}.\mathsf{pe}_n$ are embedding functions, defined below (Section 2.2.1), the $\mathsf{tf}.\mathsf{tl}^{(\ell)}$ are transformer layers, defined below (Section 2.2.2), and $\mathsf{tf}.\mathsf{out}$ is an unembedding function with some output space $\mathcal{Y}$, discussed below (Section 2.2.3).

| Function | Type | Definition / Behavior | Exact? | Reference |
|---|---|---|---|---|
| **Identity** | $\mathbb{R}^d \to \mathbb{R}^d$ | returns the input unchanged | Yes | Section 4.3 |
| **Min / Max** | $\mathbb{R}^2 \to \mathbb{R}$ | $\min(x,y)$ or $\max(x,y)$ | Yes | Section 4.4 |
| **Add / Subtract** | $\mathbb{R}^2 \to \mathbb{R}$ | $x + y$ (or $x - y$) | Yes | Section 4.5 |
| **Multiply by $c$** | $\mathbb{R} \to \mathbb{R}$ | scales input: $x \mapsto c\,x$ | Yes | Section 8.2 |
| **Multiply ($xy$)** | $\mathbb{R}^2 \to \mathbb{R}$ | product $xy$ via 2nd-order Taylor expansion of GELU (requires GELU activations) | No | Section 4.6 |
| **Comparators** | $\mathbb{R} \to \{0,1\}$ | $\mathbb{I}[x > 0]$, $\mathbb{I}[x \geq 0]$, $\mathbb{I}[x = 0]$ | No | Section 4.7 |
| **Boolean functions** | $\{0,1\}^m \to \{0,1\}$ | any Boolean function of $m$ bits | Yes | Section 4.8 |
| **Conditional (if)** | $\{0,1\} \times \mathbb{R}^2 \to \mathbb{R}$ | if $p = 1$ then $x$ else $y$ | Yes | Section 4.9 |
| **CPWL $f$** | $\mathbb{R}^d \to \mathbb{R}$ | any continuous piecewise-linear function with finitely many pieces | Yes | Section 4.1 |
| **Cancel Residual** | $\mathbb{R}^d \to \mathbb{R}^d$ | builds $f'$ so that $f'(\mathbf{x}) + \mathbf{x} = f(\mathbf{x})$ | Yes | Section 4.2 |

Table 1: Feed-forward-layer constructions. "Exact?" states whether the network realises the target function exactly or only approximately.

| Function | Definition / Behavior | Weighting | PE | Reference |
|---|---|---|---|---|
| **Identity** | passes each token through unchanged | SMAT, AHAT, UHAT | — | Section 5.1.1 |
| **Uniform Avg.** | each token outputs the mean of unmasked tokens (prefix-mean if causal mask) | SMAT, AHAT | — | Section 5.1.2 |
| **First** | retrieves the value at position 1 | SMAT, AHAT, UHAT | $(-1)^i$ | Section 5.2 |
| **Predecessor** | each token retrieves the content of position $i-1$ | AHAT, UHAT | $(-1)^i$ (or $i/n$ with *strict* mask) | Section 5.4 |
| **Index Lookup** | lookup a fixed index in the sequence | SMAT, AHAT, UHAT | see Table 5 | Section 5.3 |
| **Multi-head** | simulate multi-headed attention with one head | SMAT, AHAT, UHAT | — | Section 5.5 |
| **Tie-break** | add tiny bias $\pm 1/(j+1)$ so avg-hard emulates unique-hard | AHAT | — | Section 5.6 |
| **Hard Attn.** | simulate hard attention using soft attention and rounding/precision limits | SMAT | — | Section 5.7 |

Table 2: Self-attention layer constructions. "Weighting" lists which attention weighting function each construction works with: softmax (SMAT), average-hard (AHAT), and unique-hard (UHAT). "PE" describes the assumed positional encoding. All constructions map from $(\mathbb{R}^d)^+ \to (\mathbb{R}^d)^+$.

| Construction | Mapping | Definition / Behavior | Relevant Citations | Reference |
|---|---|---|---|---|
| **Serial Composition** | $\mathbb{R}^d \to \mathbb{R}^d$ | $f(x) = g(h(x))$ | — | Section 8.3 |
| **Parallel Composition** | $\mathbb{R}^d \to \mathbb{R}^d$ | Apply two transformers in parallel (stack the resulting output embeddings) | — | Section 8.3 |
| **Routing lemma** | $(\mathbb{R}^d)^+ \to (\mathbb{R}^d)^+$ | Reorder, duplicate, or zero out entries in hidden representations. | — | Section 8.2 |
| **Induction Head** | $\Sigma^+ \to \Sigma^+$ | If current symbol is $A$ and last occurence of $A$ was followed by $B$, then predict $B$ | Elhage et al. (2021) | Section 9.1 |
| **Dyck-1** | $\{(,)\}^+ \to \mathbb{B}$ | Recognize well-nested parentheses | Bhattamishra et al. (2020) | Section 9.2.1 |
| **Dyck-1-2** | $\{(,)\}^+ \to \mathbb{B}$ | Recognize well-nested parentheses, with maximum *nesting depth* of 2 | Yao et al. (2021), Yang et al. (2024) | Section 9.2.2 |

Table 3: Assembly and example constructions. Parallel composition requires no layernorm, architectural assumptions for Induction Head and Dyck language constructions are described in text.

Note that the above is a generalized template of the transformer architecture. Specific implementation details often differ, such as the choice of positional encoding or even activation functions. This architectural flexibility is a key strength, as it enables extensive customization while retaining key advantages, such as support for mature training and inference pipelines. We give an overview of the key components below.

### 2.2.1 Embedding

In order to process strings, the transformer needs to map them from sequences of discrete symbols to sequences of real-valued vectors. Let $x = x_1 x_2 \cdots x_n \in \Sigma^+$ be an input sequence of length $n$. In Eq. (2), each token $x_i$ is mapped to a vector $\mathsf{tf.we}(x_i) + \mathsf{tf.pe}_n(i)$, where $\mathsf{tf.we} \colon \Sigma \to \mathbb{R}^d$ is a *word embedding* and $\mathsf{tf.pe}_n \colon \mathbb{N} \to \mathbb{R}^d$ is a *position embedding*. For more on word embeddings, see Section 3.2; for more on position embeddings, see Sections 3.3 and 3.5.

### 2.2.2 Transformer Layers

Here, we will define the transformer layers and sublayers. For now, we have omitted layer normalization; it will be discussed in Section 6.

**Definition 2.2.** A *transformer layer* is a function

$$
\begin{aligned}
\mathsf{tl} \colon (\mathbb{R}^d)^+ &\xrightarrow{\text{lp}} (\mathbb{R}^d)^+ \\
\mathsf{tl}(\mathbf{z}_1, \ldots, \mathbf{z}_n) = (\mathbf{z}'_1, \ldots, \mathbf{z}'_n) \quad &\text{where} \\
(\bar{\mathbf{c}}_1, \ldots, \bar{\mathbf{c}}_n) = \mathsf{tl.sa}(\mathbf{z}_1, \ldots, \mathbf{z}_n) &+ (\mathbf{z}_1, \ldots, \mathbf{z}_n) \\
\mathbf{z}'_i = \mathsf{tl.ff}(\bar{\mathbf{c}}_i) + \bar{\mathbf{c}}_i \quad &\qquad\qquad i \in [n]
\end{aligned}
$$

where $\mathsf{tl.sa}$ is a self-attention layer (see below, Definition 2.3) and $\mathsf{tl.ff}$ is a FFN (see below, Definition 2.5).

**Self-attention** A self-attention layer computes weighted sums of *value* vectors at all positions, where the weights are determined by *query* and *key* vectors.

**Definition 2.3** (Self-Attention). A *(scaled dot-product) self-attention layer* with $d$ input/output dimensions and $d_{\text{key}}$ key/value dimensions is a length-preserving function

$$
\mathsf{sa} \colon (\mathbb{R}^d)^+ \xrightarrow{\text{lp}} (\mathbb{R}^d)^+
$$

with linear transformations

$$
\begin{aligned}
\mathsf{sa.}\mathbf{W}^{(\mathrm{Q})}, \mathsf{sa.}\mathbf{W}^{(\mathrm{K})} &\colon \mathbb{R}^d \to \mathbb{R}^{d_{\text{key}}} \\
\mathsf{sa.}\mathbf{W}^{(\mathrm{V})} &\colon \mathbb{R}^d \to \mathbb{R}^d
\end{aligned}
$$

and a weighting function

$$
\mathsf{sa.}\mathcal{S} \colon \mathbb{R}^+ \xrightarrow{\text{lp}} \mathbb{R}^+
$$

is a function where, for positions $i$ and $j$:

$$
\mathsf{sa}(\mathbf{z}_1, \ldots, \mathbf{z}_n) = (\mathbf{c}_1, \ldots, \mathbf{c}_n) \quad \text{where} \tag{6}
$$

$$
\mathbf{q}_i = \mathsf{sa.}\mathbf{W}^{(\mathrm{Q})}\,\mathbf{z}_i \tag{7}
$$

$$
\mathbf{k}_j = \mathsf{sa.}\mathbf{W}^{(\mathrm{K})}\,\mathbf{z}_j \tag{8}
$$

$$
\mathbf{v}_j = \mathsf{sa.}\mathbf{W}^{(\mathrm{V})}\,\mathbf{z}_j \tag{9}
$$

$$
s_{i,j} = \frac{\mathbf{q}_i^\top \mathbf{k}_j}{\sqrt{d_{\text{key}}}} \tag{10}
$$

$$
\alpha_{i,*} = \mathsf{sa.}\mathcal{S}(s_{i,*}) \tag{11}
$$

$$
\mathbf{c}_i = \sum_{j=1}^{n} \alpha_{i,j}\mathbf{v}_j \tag{12}
$$

We call the $s_{i,j}$ the *attention scores*, and we call the $\alpha_{i,j}$ the *attention weights*. Other variations on attention are used very often in theoretical papers; these are discussed below in Section 5.

Real transformers apply a linear transformation $\mathsf{sa}.\mathbf{W}^{(\mathrm{O})}$ to $\mathbf{c}_i$, but this does not add any expressivity, so we omit it. Real transformers also use *multi-head* self-attention, but we don't use it because it can be emulated using single-head self-attention, described in Section 5.5.

In *future-masked* (also known as *causally*-masked) self attention, every position is forced to attend only to preceding positions, by redefining the attention scores in Eq. (10) to the following, letting $\exp(-\infty) = 0$:

$$s_{i,j} = \begin{cases} \dfrac{\mathbf{q}_i^\top \mathbf{k}_j}{\sqrt{d_{\mathrm{key}}}} & j \le i \\ -\infty & \text{otherwise.} \end{cases}$$

It is sometimes convenient to consider *strictly future-masked* attention, in which case we have:

$$s_{i,j} = \begin{cases} \dfrac{\mathbf{q}_i^\top \mathbf{k}_j}{\sqrt{d_{\mathrm{key}}}} & j < i \\ -\infty & \text{otherwise.} \end{cases}$$

where it is sometimes convenient to let the first position's attention weight $\alpha_{1,1} = 0$ as a special case.

**Position-wise feed-forward networks**   A feed-forward layer computes two position-wise affine transformations, with an activation function in between. Here it is defined with ReLU, but other choices can be found in Section 4.

**Definition 2.4** (ReLU). A *rectified linear unit* or ReLU (Fukushima, 1975) is a non-linear activation function

$$\mathrm{ReLU}\colon \mathbb{R}^d \to \mathbb{R}^d$$
$$\mathrm{ReLU}(x) = \max(0, x). \tag{13}$$

**Definition 2.5** (Feed-Forward Network). A *two-layer feed-forward network* or FFN or FFNN is a function

$$\mathsf{ff}\colon \mathbb{R}^d \to \mathbb{R}^d$$
$$\mathsf{ff}(\mathbf{x}) = \mathbf{y} \quad \text{where} \tag{14}$$
$$\mathbf{h} = \mathrm{ReLU}(\mathbf{W}_1(\mathbf{x}) + \mathbf{b}_1) \tag{15}$$
$$\mathbf{y} = \mathbf{W}_2(\mathbf{h}) + \mathbf{b}_2 \tag{16}$$

with parameters

$$\mathsf{ff}.\mathbf{W}_1 \in \mathbb{R}^{d_{\mathrm{hid}} \times d} \qquad\qquad \mathsf{ff}.\mathbf{b}_1 \in \mathbb{R}^{d_{\mathrm{hid}}}$$
$$\mathsf{ff}.\mathbf{W}_2 \in \mathbb{R}^{d \times d_{\mathrm{hid}}} \qquad\qquad \mathsf{ff}.\mathbf{b}_2 \in \mathbb{R}^{d}.$$

This completes the definition of a transformer layer. In Eqs. (3) and (4), a transformer has $L$ transformer layers, each with its own parameters. After the last layer, the transformer produces an output sequence $\left(\mathbf{z}_1^{(L)}, \ldots, \mathbf{z}_n^{(L)}\right) \in (\mathbb{R}^d)^+$. There is considerable variation in what happens next.

### 2.2.3   Unembedding

After the last layer outputs a sequence of vectors $\mathbf{z}_1^{(L)}, \ldots, \mathbf{z}_n^{(L)} \in \mathbb{R}^d$, the *unembedding* function $\mathsf{tf.out}$ converts these vectors into some useful output. The most common setups are classification and language modeling.

**Classification**  In theoretical papers, it's very common to treat a transformer as a binary classifier, in order to line them up with computational complexity classes. To do this, we look only at the last output vector, $\mathbf{z}_n^{(L)}$. (Some papers, following typical usage of BERT (Devlin et al., 2019) as a classifier, add a CLS symbol to the beginning or end of the string, and use the output at that position as the classification.) This vector can be interpreted using any of the Boolean representation schemes described in Section 3.1. An extremely common choice is for tf.out to linearly project the output vector down to a scalar, and to interpret positive values as acceptance and other values as rejection:

$$\mathsf{tf}.\mathbf{W}_{\text{out}} \in \mathbb{R}^{1 \times d}$$
$$\mathbf{y}_i = \mathbb{I}\left[\mathsf{tf}.\mathbf{W}_{\text{out}}\big(\mathbf{z}_i^{(L)}\big) > 0\right]. \tag{17}$$

For multi-class classification, we can interpret $\mathbf{z}_i^{(L)}$ using any categorical representation (Section 3.2). Most commonly,

$$\mathsf{tf}.\mathbf{W}_{\text{out}} \in \mathbb{R}^{|\Sigma| \times d}$$
$$\mathbf{y}_i = \operatorname{argmax}\left(\mathsf{tf}.\mathbf{W}_{\text{out}}\big(\mathbf{z}_i^{(L)}\big)\right). \tag{18}$$

**Language modeling**  In language models, tf.out maps each $\mathbf{z}_i^{(L)}$ to a probability distribution over $\Sigma$:

$$\mathsf{tf}.\mathbf{W}_{\text{out}} \in \mathbb{R}^{|\Sigma| \times d}$$
$$\mathbf{y}_i = \operatorname{softmax}\left(\mathsf{tf}.\mathbf{W}_{\text{out}}\big(\mathbf{z}_i^{(L)}\big)\right). \tag{19}$$

This induces a probability distribution over strings $y = y_1 \cdots y_n$, given input strings $x$:

$$P(y \mid x) = \prod_{i=1}^{n} [\mathbf{y}_i]_{y_i}. \tag{20}$$

But most modern LMs are autoregressive, which means that $x_1 = \text{BOS}$, $x_i = y_{i-1}$ for $i = 2, \ldots, n$, and $y_n = \text{EOS}$. That is, the transformer takes as input a prefix of a string, and outputs a distribution over the next symbol in the string. Then Eq. (20) defines an unconditional probability distribution over strings.

## 2.3  Uniformity

In theoretical constructions of transformers, an important question is how much the construction can depend on the length of the input sequence ($n$), a concept known as uniformity. This question has practical relevance, since sequence lengths currently stretch into the millions. In this cookbook, we will observe three principles.

First, properties of a transformer not involving the parameters may depend on $n$. This includes:

- Position embeddings can be a function of $n$.

- Numerical precision of computed values can depend on $n$.

Second, the number or value of the parameters may depend on a maximum length $N$. That is, for any $N$, there exists a transformer that works on all sequences of length $n \le N$. In these cases, we will state how the number of parameters scales as a function of $N$ (e.g., $O(N)$ or $O(\log N)$).

Third, if there is any dependence on $n$ or $N$, its computational complexity should be noted (if not obvious). If unconstrained, these dependencies could allow constructions to decide undecidable languages. While such constructions can be mathematically interesting, care should be taken when drawing implications from them. Instead, it is preferable to have some constraints, such as: for each $N$, the parameter values and positional encodings for sequences of length up to $N$ can be computed in poly($\log N$) time.

| false | true | Continuous Ops. | Min. Gap | Fixed Mean | Fixed Variance |
|---|---|---|---|---|---|
| $(-\infty, 0]$ | $(0, \infty)$ | no | no | no | no |
| $0$ | $1$ | yes | yes | no | no |
| $-1$ | $+1$ | yes | yes | no | no |
| $(-\infty, 0)$ | $(0, \infty)$ | yes | no | no | no |
| $0$ | $[1, \infty)$ | yes | yes | no | no |
| $\begin{bmatrix} -1 \\ +1 \end{bmatrix}$ | $\begin{bmatrix} +1 \\ -1 \end{bmatrix}$ | yes | yes | yes | yes |
| $\begin{bmatrix} -\delta \\ +\delta \end{bmatrix}$ | $\begin{bmatrix} +\delta \\ -\delta \end{bmatrix}$ | yes | no | yes | no |

Table 4: Different Boolean representations and their properties.

## 3 Basic Ingredients

While transformers are length-preserving functions that operate on sequences of real-valued vectors, they are often used to operate on discrete data, with much freedom on how to represent these discrete values during computation. For instance, the ability to represent Boolean values or integers is often crucial for tasks like formal language recognition, string transduction, and other algorithmic constructions. Here, we discuss how to represent these discrete values in a way that is compatible with the architectural specifics of the transformer. The particular methods of representation often shed light on the expressive capacity of transformers.

### 3.1 Boolean Representations

Boolean values are a very useful and commonly used ingredient, and there are many ways to represent them (Table 4). To choose a representation, one must consider how different parts of the transformer operate on them.

**Position-wise FFNs** can be used to compute position-wise Boolean operations, covered in detail in Section 4.8, and also to convert between different representations. Since the FFNs compute continuous piecewise linear functions, this rules out representations like false $= (-\infty, 0]$, true $= (0, +\infty)$, as negation would not be a continuous function. The representations false $= 0$, true $= 1$ or false $= -1$, true $= +1$ work fine.

**Layer normalization** is covered in detail in Section 6. As discussed there (Section 6.3), we usually just want layer normalization to preserve truth values (false stays false, true stays true), so we can use a representation like

$$\text{false} = \begin{bmatrix} +1 \\ -1 \end{bmatrix} \qquad\qquad \text{true} = \begin{bmatrix} -1 \\ +1 \end{bmatrix}.$$

Unfortunately, there doesn't seem to be any one representation that has all the properties we want. It may be necessary to switch between representations as needed.

### 3.2 Categorical Representations

More generally, we may want to represent elements of a finite set. For example, word embeddings are representations of words drawn from a finite vocabulary, and position embeddings are sometimes representations of positions up to some finite maximum length. We refer to any such representation as a *categorical representation*. In theoretical constructions, categorical representations should ideally be separated by a minimum distance, for the same reasons as noted above for Booleans.

Additionally, since it's common to attend only to a particular category, the categorical representations are often orthogonal. That is, they are simply one-hot vectors. To accommodate a set of $k$ categories, a width of $d \geq k$ is required. If we are concerned about how large $d$ is, we may be able to use *almost orthogonal* vectors (Section 5.3.2).

### 3.3 Integer Representations

Integer values are another common construction, frequently used to represent counts, indices, or other discrete quantities. These representations share the same concerns as Boolean values in transformers, but furthermore the sign and magnitude of the integers can be important, as we may need to add and compare them.

Representing integers requires greater numerical precision to represent larger integers. This means that the numerical precision of the transformer plays a role in the maximum size of integers that can be represented. For more on precision, see Section 7. Since positions are integers, see also Section 3.5.

The simplest representation of an integer $C$ is just $C$ itself. But this representation is unbounded, and there is a theorem that in a transformer with bounded position embeddings and Lipschitz position-wise functions (Hahn, 2020) or even non-Lipschitz layer normalization (Chiang et al., 2023), all computed values are bounded. So it may be difficult to compute values in this representation.

Instead, it's very common to see a count $C$ stored as $C/n$ where $n$ is the length of the input (Chiang et al., 2023) or $C/i$ where $i$ is the position where this integer is stored. We often end up with the former when using unmasked uniform attention to compute integer values, and the latter when using future-masked uniform attention. A scale-invariant representation of integers is via the layer norm (Section 5.3.3).

Since each position has the same denominator, position-wise operations are straightforward to implement using this representation. Position-wise addition and subtraction is described in Section 4.5. Position-wise comparison of integers is described—with some important caveats—in Section 4.7.

### 3.4 Special Symbols

In Section 2.2.3, we mentioned the special symbols BOS (beginning of sequence), EOS (end of sequence), and CLS (classification). Having one of these tokens in the sequence also allows a transformer to compute the value $\frac{1}{n+1}$ (or $\frac{1}{i+1}$ if future-masked) by using uniform attention (Section 5.1.2).

### 3.5 Positional Encodings

Transformers use positional encodings as a method for incorporating information about token positions, since neither self-attention nor position-wise FFNs have an inherent representation of token ordering. The original transformer model (Vaswani et al., 2017) used sinusoidal positional encodings, but since then, many encodings have been proposed, addressing length generalization (Kazemnejad et al., 2023), relative distances between tokens (Shaw et al., 2018), as well as representing tree structure in sequences (Shiv & Quirk, 2019). In this section, we focus on different kinds of positional encodings and how they may affect the expressiveness of the model. Here we present *scalar* positional encodings, according to the definition of the token embedding function Section 2.2.1, a positional encoding is a vector in $\mathbb{R}^d$. It is common to choose a single positional encoding (i.e. a vector in $\mathbb{R}^1$) however some constructions make use of multiple positional encodings in different dimensions of the emebdding.

#### 3.5.1 Simple Positional Encodings

**The value** $\frac{1}{i}$   Obtaining the value $\frac{1}{i}$ at position $i$ can be introduced via a positional encoding, or it can be computed using a future-masked uniform attention layer (Section 5.1.2) and the presence of a BOS token. This positional encoding plays a role in the constructions of Barcelo et al. (2024), Merrill & Sabharwal (2024a), and others. Additionally, as described in Section 5.6, this value may be used to simulate a UHAT using an AHAT.

**Length-averaged** $\frac{i}{n}$   Similar to above, the value $\frac{i}{n}$ may also be obtained at position $i$ using an unmasked uniform attention layer Section 5.1.2 and the presence of a beginning-of-sequence token. This positional encoding can be found in the constructions of Merrill & Sabharwal (2023); Chiang & Cholak (2022); Strobl et al. (2025). One reason for using $\frac{i}{n}$ in place of $i$ is that sometimes it's desirable for the position encoding to be bounded.

**Powers of** $i$   Powers of $i$ other than $i^{-1}$ can also be used as positional encodings. In particular, $i$ and $i^2$ are crucial to perform certain index lookups in Section 5.3. Higher powers like $i^3$ are used by Yang et al. (2026) in order to simulate table-lookup exactly using soft-attention. Generally, these large powers of $i$ are used to create attention scores that scale rapidly with the sequence length, to ensure that attention can be focused on a single position.

### 3.5.2   Sinusoidal Positional Encoding

The original paper on transformers (Vaswani et al., 2017) used *sinusoidal position encodings*. Suppose that the embedding dimension $d$ is even, then for $0 \le c \le \frac{d}{2} - 1$, let:

$$\mathsf{pe}(i, 2c + 1) = \sin\left(\frac{i}{M^{2c/d}}\right)$$

$$\mathsf{pe}(i, 2c) = \cos\left(\frac{i}{M^{2c/d}}\right)$$

where $M$ is a large number, such as $M = 10000$ (Vaswani et al., 2017). Sinusoidal positional encodings. Using the table-lookup operation described in Section 5.3, these positional encodings can also be used to do modulo counting – to recognize PARITY, for instance.

## 4   Feed-Forward Layers

Feed-forward layers are a fundamental building block of transformers. Empirically, they have been observed to contribute to the expressive power of transformers (Geva et al., 2021). Here, we show how feed-forward layers can compute a variety of important functions. Given enough parameters and a large enough input and hidden dimension, a feed-forward layer can approximate a very large class of functions (Cybenko, 1989; Hornik et al., 1989), but here we focus on those that can be expressed exactly, with a fixed number of parameters.

Much of the expressive power of feed-forward layers comes from their non-linear *activation functions*. The concept was originally inspired by the threshold-based firing of biological neurons (McCulloch & Pitts, 1943), and they were later shown to be essential for allowing neural networks to learn non-linear functions, a critical property for modeling complex data (Hornik et al., 1989). We focus here on two popular activation functions, the rectified linear unit (ReLU) and Gaussian error linear unit (GELU). Other notable activation functions include sigmoid, softmax, hyperbolic tangent, exponential linear unit (ELU), and gated linear unit (GLU).

Recall (Definition 2.5) that a FFN has the form

$$\mathsf{FFN}(\mathbf{x}) = \mathbf{W}_2 \mathrm{ReLU}(\mathbf{W}_1 \mathbf{x} + \mathbf{b}_1) + \mathbf{b}_2.$$

In the following constructions, we will specify for each FFN the parameters $\mathbf{W}_1, \mathbf{W}_2, \mathbf{b}_1$, and $\mathbf{b}_2$, which can be plugged into Definition 2.5.

### 4.1   Continuous Piecewise Linear Functions

ReLU FNNs may be used to represent any continuous piecewise linear function made up of a finite number of linear segments (pieces). We first define such functions as follows.

**Definition 4.1** (Continuous Piecewise Linear Function)**.** Let $X \subseteq \mathbb{R}^d$. A function $f : X \to \mathbb{R}$ is *continuous piecewise linear (CPWL)* if there are closed polyhedral subsets $X_1, \ldots, X_n \subseteq X$ such that $\bigcup_{i \in [n]} X_i = X$, and for all $i \in [n]$, $f|_{X_i}$ is affine.

For the univariate case ($d = 1$), let $f\colon \mathbb{R} \to \mathbb{R}$ be CPWL. Assume that $f$ has $n \geq 2$ pieces (of which the first and last extend to infinity) and is represented by $x_1 < \ldots < x_{n+1}$ and $y_1, \ldots, y_{n+1}$ such that $f(x_k) = y_k$ for $k = 1, \ldots, n+1$. Points $(x_2, y_2)$ to $(x_n, y_n)$ are the "knots" of $f$, while points $(x_1, y_1)$ and $(x_{n+1}, y_{n+1})$ lie in the first and last piece.

The main idea is to concatenate a sequence of ReLU components at these knots, such that each component approximates one linear piece of $f$ while "undoing" the effect of its immediate predecessor. Specifically, we first rewrite $f$ as:

$$f(x) = y_1 + m_1(x - x_1) + \sum_{k=2}^{n} (m_k - m_{k-1}) \operatorname{ReLU}(x - x_k)$$

$$= \begin{cases} y_1 + m_1(x - x_1), & x \leq x_2 \\ y_k + m_k(x - x_k), & x_k \leq x \leq x_{k+1} \\ y_n + m_n(x - x_n), & x \geq x_n. \end{cases}$$

where $m_1, \ldots, m_n$ are the slopes defined as:

$$m_k = \frac{y_{k+1} - y_k}{x_{k+1} - x_k}, \quad \text{for } k = 1, \ldots, n.$$

This is achievable with the following FFN of hidden dimension $n + 1$:

$$f.\mathbf{W}_1 = \left.\begin{bmatrix} -1 \\ 1 \\ 1 \\ 1 \\ \vdots \\ 1 \end{bmatrix}\right\} n \text{ copies} \qquad f.\mathbf{b}_1 = \begin{bmatrix} 0 \\ 0 \\ -x_2 \\ -x_3 \\ \vdots \\ -x_n \end{bmatrix}$$

$$f.\mathbf{W}_2 = \begin{bmatrix} -m_1 & m_1 & (m_2 - m_1) & \cdots & (m_n - m_{n-1}) \end{bmatrix} \qquad f.\mathbf{b}_2 = \begin{bmatrix} y_1 - m_1 x_1 \end{bmatrix}.$$

For the multivariate case ($f\colon \mathbb{R}^d \to \mathbb{R}$), any CPWL function with $k$ pieces can be computed exactly by a FFN with $O(\log k)$ layers (Arora et al., 2018); we don't reproduce this construction here, but below, we show some cases that can be computed in two layers.

## 4.2 Canceling Residual Connections

As noted above, an FFN $f\colon \mathbb{R}^d \to \mathbb{R}^d$ is normally used with a residual connection, $\mathbf{y} = f(\mathbf{x}) + \mathbf{x}$. Sometimes, we don't want the residual connection, and fortunately, it's always possible to cancel it out (Chiang et al., 2023). That is, there is an FFN $f'$ with parameters

$$f'.\mathbf{W}_1 = \begin{bmatrix} f.\mathbf{W}_1 \\ \mathbf{I} \\ -\mathbf{I} \end{bmatrix} \qquad f'.\mathbf{b}_1 = \begin{bmatrix} f.\mathbf{b}_1 \\ \mathbf{0} \\ \mathbf{0} \end{bmatrix}$$

$$f'.\mathbf{W}_2 = \begin{bmatrix} f.\mathbf{W}_2 & -\mathbf{I} & \mathbf{I} \end{bmatrix} \qquad f'.\mathbf{b}_2 = f.\mathbf{b}_2$$

so that $\mathbf{y} = f'(\mathbf{x}) + \mathbf{x} = f(\mathbf{x})$; that is, $f'$ with a residual connection behaves like $f$ without a residual connection. From now on, we assume without loss of generality that residual connections are optional.

## 4.3 Identity Function

We very often need an FFN that does nothing, either because we need two self-attentions in a row or as a building block for the other constructions below.

$$\mathsf{id}\colon \mathbb{R}^d \to \mathbb{R}^d$$

$$\mathsf{id}(x) = x.$$

If a residual connection is present, then this is straightforward: zero out the FFN and retain only the residual connection. In particular, the zero FNN is defined as follows:

$$\mathsf{zero}.\mathbf{W}_1 = \mathsf{zero}.\mathbf{W}_2 = \mathbf{0}_{d \times d}$$
$$\mathsf{zero}.\mathbf{b}_1 = \mathsf{zero}.\mathbf{b}_2 = \mathbf{0}_d,$$

Then, when combined with a residual connection, the zero FNN satisfies the identity $x + \mathsf{zero}(x) = x$.

If a residual connection is not present, however, then some encoding is necessary. We first consider the simplified case of embedding dimension $d = 1$, in which case we set:

$$\mathsf{id}.\mathbf{W}_1 = \begin{bmatrix} 1 \\ -1 \end{bmatrix} \qquad \mathsf{id}.\mathbf{b}_1 = \mathbf{0}$$

$$\mathsf{id}.\mathbf{W}_2 = \begin{bmatrix} 1 & -1 \end{bmatrix} \quad \mathsf{id}.\mathbf{b}_2 = 0. \tag{21}$$

This constructs the identity FFN, as it would expand as:

$$\mathsf{id}(x) = \mathrm{ReLU}(x) + -\mathrm{ReLU}(-x) = x.$$

To generalize this to vectors in $\mathbb{R}^d$ ($d \geq 1$), we may use parallel composition (Lemma 8.3) and routing (Lemma 8.1) to stack $d$ copies of this construction into a single FFN:

$$\mathsf{id}.\mathbf{W}_1 = \begin{bmatrix} \mathbf{I}^d \\ -\mathbf{I}^d \end{bmatrix} \qquad \mathsf{id}.\mathbf{b}_1 = \mathbf{0}_{2d}$$

$$\mathsf{id}.\mathbf{W}_2 = \begin{bmatrix} \mathbf{I}^d & -\mathbf{I}^d \end{bmatrix} \quad \mathsf{id}.\mathbf{b}_2 = \mathbf{0}_d \tag{22}$$

In general, we will state constructions using scalars when vectors are not necessary, but they can be generalized to vectors as we have done here.

### 4.4 Min and Max

It is often desirable to compute the minimum or maximum of two numbers:

$$\mathsf{min}, \mathsf{max} \colon \mathbb{R} \times \mathbb{R} \to \mathbb{R}$$

$$\mathsf{min}\left( \begin{bmatrix} x \\ y \end{bmatrix} \right) = \min(x, y)$$

$$\mathsf{max}\left( \begin{bmatrix} x \\ y \end{bmatrix} \right) = \max(x, y).$$

These functions are CPWL, so there exist FFNs to compute them:

$$\mathsf{min}.\mathbf{W}_1 = \begin{bmatrix} 1 & 0 \\ -1 & 0 \\ 1 & -1 \end{bmatrix} \qquad\qquad \mathsf{min}.\mathbf{b}_1 = \mathbf{0}$$

$$\mathsf{min}.\mathbf{W}_2 = \begin{bmatrix} 1 & -1 & -1 \end{bmatrix} \qquad\qquad \mathsf{min}.\mathbf{b}_2 = \mathbf{0}$$

$$\mathsf{max}.\mathbf{W}_1 = \begin{bmatrix} 1 & 0 \\ -1 & 0 \\ -1 & 1 \end{bmatrix} \qquad\qquad \mathsf{max}.\mathbf{b}_1 = \mathbf{0}$$

$$\mathsf{max}.\mathbf{W}_2 = \begin{bmatrix} 1 & -1 & 1 \end{bmatrix} \qquad\qquad \mathsf{max}.\mathbf{b}_2 = \mathbf{0}$$

Then

$$\mathsf{min}\left( \begin{bmatrix} x \\ y \end{bmatrix} \right) = \mathrm{ReLU}(x) - \mathrm{ReLU}(-x) - \mathrm{ReLU}(x - y) = x - \mathrm{ReLU}(x - y) = \min(x, y)$$

$$\mathsf{max}\left( \begin{bmatrix} x \\ y \end{bmatrix} \right) = \mathrm{ReLU}(x) - \mathrm{ReLU}(-x) + \mathrm{ReLU}(y - x) = x + \mathrm{ReLU}(y - x) = \max(x, y).$$

Note that this construction maps from $\mathbb{R}^2$ to $\mathbb{R}^1$. If one wishes to make the input-output dimensionalities the same, then we may pad the output weights with zero-valued rows. For example, to make the output of a max component in $\mathbb{R}^2$, we may zero-pad the output weights to shape $\mathsf{max}.\mathbf{W}_2 \in \mathbb{R}^{2\times 3}$ and $\mathsf{max}.\mathbf{b}_2 \in \mathbb{R}^2$. Similar padding ideas can be applied to other constructions to enforce desired dimensionalities.

### 4.5 Addition and Subtraction

Addition (or subtraction, or any linear function) is an easy extension of the identity (Eq. (21)).

$$\mathsf{add}\colon \mathbb{R} \times \mathbb{R} \to \mathbb{R}$$
$$\mathsf{add}(x, y) = x + y.$$

$$\mathsf{add}.\mathbf{W}_1 = \begin{bmatrix} 1 & 0 \\ -1 & 0 \\ 0 & 1 \\ 0 & -1 \end{bmatrix} \qquad \mathsf{add}.\mathbf{b}_1 = \mathbf{0}$$

$$\mathsf{add}.\mathbf{W}_2 = \begin{bmatrix} 1 & -1 & 1 & -1 \end{bmatrix} \qquad \mathsf{add}.\mathbf{b}_2 = \mathbf{0}$$

Then

$$\mathsf{add}\left(\begin{bmatrix} x \\ y \end{bmatrix}\right) = \mathrm{ReLU}(x) - \mathrm{ReLU}(-x) + \mathrm{ReLU}(y) - \mathrm{ReLU}(-y) = x + y.$$

### 4.6 Multiplication

Multiplication by a constant $c$ is easy (use the identity recipe (Section 4.3) with the routing lemma (Lemma 8.1) to premultiply or postmultiply by $c$), but multiplication of two activations can only be approximated, and requires an activation function with nonzero second derivative (Akyürek et al., 2023, Lemma 4). Feng et al. (2023, Lemma C.1) give a similar approximation.

$$\mathsf{mul}\colon \mathbb{R} \times \mathbb{R} \to \mathbb{R}$$
$$\mathsf{mul}\left(\begin{bmatrix} x \\ y \end{bmatrix}\right) \approx xy.$$

What we ideally want is a quadratic activation function. Here, we use the Gaussian Error Linear Unit (GELU), which is used in modern transformer architectures like BERT (Devlin et al., 2019) and GPT (Radford et al., 2018). It is not exactly quadratic, but we can think of it as an approximation of its second-order Taylor approximation, which is.

**Definition 4.2** (GELU). A *Gaussian error linear unit* or GELU (Hendrycks & Gimpel, 2016) is a non-linear activation function

$$\mathrm{GELU}\colon \mathbb{R}^d \to \mathbb{R}^d$$
$$\mathrm{GELU}(x) = x\,\Phi(x) \tag{23}$$
$$= \frac{x}{2}\left(1 + \mathrm{erf}\left(\frac{x}{\sqrt{2}}\right)\right) \tag{24}$$
$$\approx \frac{x}{2}\left(1 + \tanh\left(\sqrt{\frac{2}{\pi}}\left(x + 0.044715x^3\right)\right)\right) \tag{25}$$
$$\approx x\,\mathrm{sigmoid}(1.702x) \tag{26}$$

where $\Phi$ is the cumulative distribution function of the standard normal distribution, erf is the Gauss error function, and $\mathrm{sigmoid}(x) = 1/(1 + e^{-x})$.

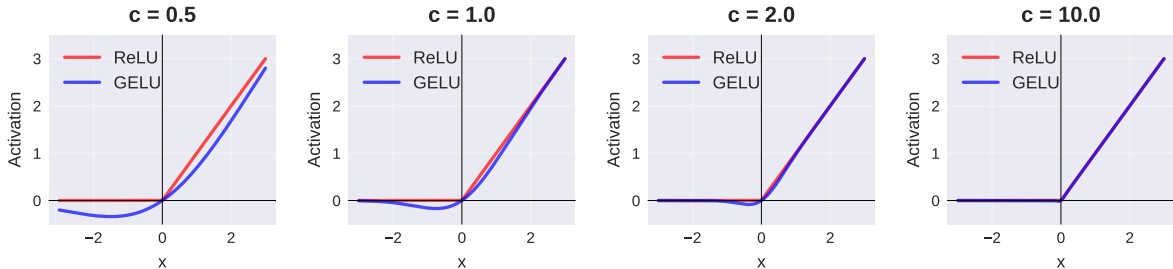

Figure 1: $c^{-1}\text{GELU}(cx) \approx \text{ReLU}(x)$ for sufficiently large $c > 0$. Many of our ReLU-based constructions can thus be implemented with GELUs given appropriate scaling and error analysis near the origin.

The approximations (25) and (26) are from Hendrycks & Gimpel (2016). PyTorch implements a choice between (24) or (25).[1]. While GELU is a popular activation in real-world transformer implementations, ReLU remains popular for theoretical analysis as it is a simpler function. In particular, it is possible to use GELU to approximate ReLU by noting that $c^{-1}\text{GELU}(cx) \approx \text{ReLU}(x)$ for sufficiently large $c > 0$, see Fig. 1, which occurs because:

$$c^{-1}\text{GELU}(cx) = c^{-1}cx\Phi(cx) = x\Phi(cx), \tag{27}$$

where note that the scaled Gaussian CDF $\Phi(cx)$ approaches a discrete step function in the limit:

$$\lim_{c \to \infty} \Phi(cx) = \begin{cases} 1, & x \geq 0, \\ 0, & x < 0. \end{cases} \tag{28}$$

Regardless of whether GELU is defined as Eq. (24) or Eq. (25), we have

$$\text{GELU}(z) = \frac{z}{2} + \frac{z^2}{\sqrt{2\pi}} + R(z)$$

where the Lagrange remainder term is, for some $\xi \in [0, z]$,

$$R(z) = \frac{1}{6}\text{GELU}'''(\xi)z^3$$

$$|R(z)| \leq \frac{1}{6}|z|^3.$$

From this, we can derive

$$\sqrt{\frac{\pi}{2}}(\text{GELU}(x + y) - \text{GELU}(x) - \text{GELU}(y)) = xy + \epsilon(x, y)$$

where the error is

$$|\epsilon(x, y)| \leq \frac{1}{4}(|x| + |y|)^3.$$

Thus we can construct an FFN with GELU activations (implemented using the first two terms fo the Taylor expansion) and the following parameters:

$$\text{mul}.\mathbf{W}_1 = \begin{bmatrix} 1 & 1 \\ 1 & 0 \\ 0 & 1 \end{bmatrix} \qquad\qquad \text{mul}.\mathbf{b}_1 = \mathbf{0}$$

$$\text{mul}.\mathbf{W}_2 = \left[\sqrt{\frac{\pi}{2}} \quad -\sqrt{\frac{\pi}{2}} \quad -\sqrt{\frac{\pi}{2}}\right] \qquad\qquad \text{mul}.\mathbf{b}_2 = \mathbf{0}.$$

[1] https://pytorch.org/docs/stable/generated/torch.nn.GELU.html

### 4.7 Comparisons

Binary-valued comparisons ($=$, $<$, $\leq$, etc.) are not CPWL because there is a jump from 0 to 1. However, we may approximate them using FFNs. If some position-independent and length-independent tolerance $\epsilon$ is known apriori, we may construct these FFNs as follows, where below we specify the intended behavior, show its implementation as FFN weights, and give a visualization plot.

$$\text{GTZero}_\epsilon(x) = \begin{cases} 0 & x \leq 0 \\ x/\epsilon & 0 < x < \epsilon \\ 1 & \epsilon \leq x \end{cases}$$

$$\text{GTZero}_\epsilon.\mathbf{W}_1 = \begin{bmatrix} 1 \\ 1 \end{bmatrix} \qquad \text{GTZero}_\epsilon.\mathbf{b}_1 = \begin{bmatrix} 0 \\ -\epsilon \end{bmatrix}$$

$$\text{GTZero}_\epsilon.\mathbf{W}_2 = \begin{bmatrix} 1/\epsilon & -1/\epsilon \end{bmatrix} \qquad \text{GTZero}_\epsilon.\mathbf{b}_2 = 0$$

(29)

$$\text{GEZero}_\epsilon(x) = \begin{cases} 0 & x \leq -\epsilon \\ 1 + (x/\epsilon) & -\epsilon < x < 0 \\ 1 & 0 \leq x \end{cases}$$

$$\text{GEZero}_\epsilon.\mathbf{W}_1 = \begin{bmatrix} 1 \\ 1 \end{bmatrix} \qquad \text{GEZero}_\epsilon.\mathbf{b}_1 = \begin{bmatrix} \epsilon \\ 0 \end{bmatrix}$$

$$\text{GEZero}_\epsilon.\mathbf{W}_2 = \begin{bmatrix} 1/\epsilon & -1/\epsilon \end{bmatrix} \qquad \text{GEZero}_\epsilon.\mathbf{b}_2 = 0$$

(30)

$$\text{EqZero}_\epsilon(x) = \begin{cases} 0 & |x| \geq \epsilon \\ 1 - |x/\epsilon| & 0 < |x| < \epsilon \\ 1 & x = 0 \end{cases}$$

$$\text{EqZero}_\epsilon.\mathbf{W}_1 = \begin{bmatrix} 1 \\ 1 \\ 1 \end{bmatrix} \qquad \text{EqZero}_\epsilon.\mathbf{b}_1 = \begin{bmatrix} \epsilon \\ 0 \\ -\epsilon \end{bmatrix}$$

$$\text{EqZero}_\epsilon.\mathbf{W}_2 = \begin{bmatrix} 1/\epsilon & -2/\epsilon & 1/\epsilon \end{bmatrix} \qquad \text{EqZero}_\epsilon.\mathbf{b}_2 = 0$$

(31)

We emphasize that the above constructions compute exact comparisons only for some inputs. Inputs where the construction does not work as intended are shown in gray and are to be avoided.

Alternatively, if the desired threshold $\epsilon$ is input-dependent, and we are also willing to accept values as small as $x \geq \epsilon$ as "true", then we may use the following $\epsilon$-parameterized constructions:

$$\text{GTZero}(x, \epsilon) = \begin{cases} 0 & x \le 0 \\ x & 0 < x < \epsilon \\ \epsilon & \epsilon \le x \end{cases}$$

$$\text{GTZero}.\mathbf{W}_1 = \begin{bmatrix} 1 & 0 \\ 1 & -1 \end{bmatrix} \qquad \text{GTZero}_\epsilon.\mathbf{b}_1 = \mathbf{0}$$

$$\text{GTZero}_\epsilon.\mathbf{W}_2 = \begin{bmatrix} 1 & -1 \end{bmatrix} \qquad \text{GTZero}_\epsilon.\mathbf{b}_2 = 0 \tag{32}$$

$$\text{GEZero}(x, \epsilon) = \begin{cases} 0 & x \le -\epsilon \\ x + \epsilon & -\epsilon < x < 0 \\ \epsilon & 0 \le x \end{cases}$$

$$\text{GEZero}.\mathbf{W}_1 = \begin{bmatrix} 1 & 1 \\ 1 & 0 \end{bmatrix} \qquad \text{GEZero}.\mathbf{b}_1 = \mathbf{0}$$

$$\text{GEZero}.\mathbf{W}_2 = \begin{bmatrix} 1 & -1 \end{bmatrix} \qquad \text{GEZero}.\mathbf{b}_2 = 0 \tag{33}$$

$$\text{EqZero}(x, \epsilon) = \begin{cases} 0 & |x| \ge \epsilon \\ \epsilon - |x| & 0 < |x| < \epsilon \\ \epsilon & x = 0 \end{cases}$$

$$\text{EqZero}.\mathbf{W}_1 = \begin{bmatrix} 1 & 1 \\ 1 & 0 \\ 1 & -1 \end{bmatrix} \qquad \text{EqZero}.\mathbf{b}_1 = \mathbf{0}$$

$$\text{EqZero}.\mathbf{W}_2 = \begin{bmatrix} 1 & -2 & 1 \end{bmatrix} \qquad \text{EqZero}.\mathbf{b}_2 = 0 \tag{34}$$

Similar to the input-independent constructions, input values where the computation is not exact are shown in gray and are to be avoided.

## 4.8 Boolean Functions

In this section, we show how to compute arbitrary Boolean functions using a single feed-forward network. We show the construction for true $= 1$, false $= 0$ (see Section 3.1). This is probably the easiest case, but the others are not much more difficult.

If we are not concerned about depth, the connectives $\wedge$, $\vee$, $\neg$ can be computed by FFNs with ReLU activations. Conjunction ($\wedge$) is equivalent to min, disjunction ($\vee$) is equivalent to max, and logical negation ($\neg$) is just $(1 - x)$.

But we can also pack an arbitrary Boolean function $\phi \colon \{0, 1\}^m \to \{0, 1\}$ into a single two-layer FFN. The easiest (not necessarily the most efficient) way to do this is to list out all the possible inputs and outputs of $f$. There are $2^m$ possible truth assignments to the variables $x_1, \ldots, x_m$; number them $\xi_0, \xi_1, \ldots, \xi_{2^m-1} \in \{0, 1\}^m$. That is, $[\xi_k]_i$ is the $i$-th bit of $k$, and $\mathbf{1} \cdot \xi_k$ is the number of variables that $\xi_k$ makes true.

$$\text{ff}_\phi.\mathbf{W}_1 = \begin{bmatrix} (2\xi_0 - \mathbf{1})^\top \\ (2\xi_1 - \mathbf{1})^\top \\ (2\xi_2 - \mathbf{1})^\top \\ \vdots \\ (2\xi_{2^m-1} - \mathbf{1})^\top \end{bmatrix} = \begin{bmatrix} -1 & \cdots & -1 & -1 \\ -1 & \cdots & -1 & 1 \\ -1 & \cdots & 1 & -1 \\ \vdots & \ddots & \vdots & \vdots \\ 1 & \cdots & 1 & 1 \end{bmatrix} \quad \text{ff}_\phi.\mathbf{b}_1 = \begin{bmatrix} -\mathbf{1} \cdot \xi_0 + 1 \\ -\mathbf{1} \cdot \xi_1 + 1 \\ -\mathbf{1} \cdot \xi_2 + 1 \\ \vdots \\ -\mathbf{1} \cdot \xi_{2^m-1} + 1 \end{bmatrix} = \begin{bmatrix} 1 \\ 0 \\ 0 \\ \vdots \\ -m + 1 \end{bmatrix}$$

$$\text{ff}_\phi.\mathbf{W}_2 = \begin{bmatrix} \phi(\xi_0) & \cdots & \phi(\xi_{2^m-1}) \end{bmatrix} \qquad \text{ff}_\phi.\mathbf{b}_2 = \mathbf{0}. \tag{35}$$

We want $h_k$ to test, for each truth assignment $\xi_k$, where $\mathbf{x} = \xi_k$. Consider the vectors $(2\xi_k - \mathbf{1})$ and $(2\mathbf{x} - \mathbf{1})$, whose entries are all $\pm 1$: they are equal if and only iff their dot-product is $m$. So we want

$$
\begin{aligned}
h_k &= \mathrm{ReLU}\left( \begin{bmatrix} \frac{1}{2}(2\xi_0 - \mathbf{1}) \cdot (2\mathbf{x} - \mathbf{1}) - (\frac{m}{2} - 1) \\ \vdots \\ \frac{1}{2}(2\xi_{2^m - 1} - \mathbf{1}) \cdot (2\mathbf{x} - \mathbf{1}) - (\frac{m}{2} - 1) \end{bmatrix} \right) \\
&= \mathrm{ReLU}\left( \begin{bmatrix} 2\xi_0 \cdot \mathbf{x} - \mathbf{x} \cdot \mathbf{1} - \xi_0 \cdot \mathbf{1} + 1 \\ \vdots \\ 2\xi_{2^m - 1} \cdot \mathbf{x} - \mathbf{x} \cdot \mathbf{1} - \xi_{2^m - 1} \cdot \mathbf{1} + 1 \end{bmatrix} \right) \\
&= \mathrm{ReLU}\left( \begin{bmatrix} (2\xi_0 - \mathbf{1}) \cdot \mathbf{x} - \mathbf{1} \cdot \xi_0 + 1 \\ \vdots \\ (2\xi_{2^m - 1} - \mathbf{1}) \cdot \mathbf{x} - \mathbf{1} \cdot \xi_{2^m - 1} + 1 \end{bmatrix} \right) \\
&= \mathrm{ReLU}(\mathsf{ff}_\phi.\mathbf{W}_1(\mathbf{x}) + \mathsf{ff}_\phi.\mathbf{b}_1).
\end{aligned}
$$

Then the second layer tests if any truth assignment $\xi_k$ makes $\phi$ true:

$$
\begin{aligned}
y &= \sum_k \phi(\xi_k)\, h_k \\
&= \sum_k \phi(\xi_k)\, \mathbb{I}\,[\xi_k = \mathbf{x}] \\
&= \phi(\mathbf{x}).
\end{aligned}
$$

### 4.9 Conditionals

Suppose we want to compute the conditional expression

$$
\mathsf{if}(p, x, y) = \begin{cases} x & \text{if } p = 1 \\ y & \text{if } p = 0. \end{cases}
$$

We assume that $x, y \in [0, 1]$, but the construction is easily generalized for any bounded interval. We adapt a construction used by Pérez et al. (2021, Lemma 11) and Merrill & Sabharwal (2024a, Theorem 1).

$$
\mathsf{if}.\mathbf{W}_1 = \begin{bmatrix} 1 & 1 & 0 \\ -1 & 0 & 1 \end{bmatrix} \qquad\qquad \mathsf{if}.\mathbf{b}_1 = \begin{bmatrix} -1 \\ 0 \end{bmatrix}
$$

$$
\mathsf{if}.\mathbf{W}_2 = \begin{bmatrix} 1 & 1 \end{bmatrix} \qquad\qquad \mathsf{if}.\mathbf{b}_2 = 0.
$$

Then

$$
\mathsf{if}\left( \begin{bmatrix} 1 \\ x \\ y \end{bmatrix} \right) = \mathrm{ReLU}(x) + \mathrm{ReLU}(y - 1) = x
$$

$$
\mathsf{if}\left( \begin{bmatrix} 0 \\ x \\ y \end{bmatrix} \right) = \mathrm{ReLU}(x - 1) + \mathrm{ReLU}(y) = y.
$$

## 5   Self-Attention Layers

Attention layers are the fundamental ingredient of transformers, allowing computations across positions in the sequence in a parallelizable manner (Vaswani et al., 2017). The original motivation for self-attention was to compute the relationships between source and target words in machine translation (Bahdanau et al., 2015), but since then the mechanism has been trained to perform a huge variety of different tasks. In this

section we primarily explain how attention can be used to retrieve information from different positions in specific ways.

While attention layers were defined in Definition 2.3, there are additional design choices that can be made for the ease of implementing particular constructions, which we detail below.

**Attention Masking**   While transformers are typically implemented using future-masked attention or with no masking, in *past-masked attention*, we have

$$
s_{ij} = \begin{cases} \dfrac{\mathbf{q}_i^\top \mathbf{k}_j}{\sqrt{d_{\text{key}}}} & j \geq i \\ -\infty & \text{otherwise.} \end{cases}
$$

**Weighting Function**   The weighting function $\mathcal{S}\colon \mathbb{R}^+ \xrightarrow{\text{lp}} \mathbb{R}^+$ computes the attention weights $\alpha_{i,*}$ based on the attention scores $s_{i,*}$. A common choice is the *softmax* function:

$$
[\operatorname{softmax}(s_1, \ldots, s_n)]_j = \frac{e^{s_j}}{\sum_{k=1}^{n} e^{s_k}}.
$$

But we consider several alternatives below.

**Hard Attention**   In hard attention, the attention weights are assigned to focus all attention on the maximum-scoring position or positions.

**Definition 5.1** (Hardmax). The leftmost, rightmost, and average-hardmax functions are defined as follows. For any sequence of scores $\mathbf{s} = (s_1, \ldots, s_n)$, let

$$
I(\mathbf{s}) = \{i \in [|\mathbf{s}|] \mid s_i = \max \mathbf{s}\}
$$

be the set of indices of maximal scores. The lhardmax and rhardmax functions return a one-hot vector with a 1 at the position of the leftmost or rightmost maximal element, respectively:

$$
[\operatorname{lhardmax}(\mathbf{s})]_i = \mathbb{I}[i = \min I(\mathbf{s})]
$$
$$
[\operatorname{rhardmax}(\mathbf{s})]_i = \mathbb{I}[i = \max I(\mathbf{s})].
$$

The ahardmax function pays equal attention to all the maximal elements:

$$
[\operatorname{ahardmax}(\mathbf{s})]_i = \frac{1}{|I(\mathbf{s})|} \mathbb{I}[i \in I(\mathbf{s})].
$$

Leftmost-hard attention was previously called *hard* attention by Hahn (2020) and *unique-hard* attention by Hao et al. (2022). One may also consider rightmost-hard attention, in which the rightmost maximal element is used. Average-hard attention was also called *hard* attention by Pérez et al. (2021) and *saturated* attention by Merrill et al. (2022), and has been argued to be a realistic approximation to how trained transformers behave in practice (Merrill et al., 2021). Neither type of hard attention should be confused with the concept of hard attention used in computer vision (e.g., Xu et al., 2015).

## 5.1   Trivial Cases

These trivial cases are the most basic ways an attention layer can process sequence-wise information: leave it unchanged, or aggregate them uniformly.

### 5.1.1   Identity

We start with the identity function,

$$
\mathsf{id}\colon \mathbb{R}^+ \xrightarrow{\text{lp}} \mathbb{R}^+
$$
$$
\mathsf{id}(\mathbf{x}) = \mathbf{x}.
$$

As with FFNNs (Section 4.3), the easiest way to compute the identity function is to use the residual connection, by setting the value vectors to zero:

$$\text{att\_zero.}\mathbf{W}^{(Q)} = \mathbf{0} \qquad \text{att\_zero.}\mathbf{W}^{(K)} = \mathbf{0} \qquad \text{att\_zero.}\mathbf{W}^{(V)} = \mathbf{0}$$

Then, the residual connection will pass the input through unchanged.

$$\text{id}(\mathbf{x}) = \text{att\_zero}(\mathbf{x}) + \mathbf{x}.$$

### 5.1.2  Average

We can use attention to implement averaging of vectors or scalars, as follows:

$$\text{Avg}, \text{Avg}_{\leftarrow} : (\mathbb{R}^d)^+ \xrightarrow{\text{lp}} (\mathbb{R}^d)^+$$

$$\text{Avg}(\mathbf{x}_1, \mathbf{x}_2, \ldots, \mathbf{x}_n)_i = \frac{1}{n} \sum_{k=1}^{n} \mathbf{x}_k \quad \text{for } i \in [N]$$

$$\text{Avg}_{\leftarrow}(\mathbf{x}_1, \mathbf{x}_2, \ldots, \mathbf{x}_n)_i = \frac{1}{i} \sum_{k=1}^{i} \mathbf{x}_k \quad \text{for } i \in [N]$$

We can make each position attend uniformly to all (unmasked) positions by setting the query and key matrices to zero.

$$\text{Avg.}\mathbf{W}^{(Q)} = \mathbf{0} \qquad \text{Avg.}\mathbf{W}^{(K)} = \mathbf{0} \qquad \text{Avg.}\mathbf{W}^{(V)} = \mathbf{I}$$

And $\text{Avg}_{\leftarrow}$ is defined the same way, but uses future masking. For $\text{Avg}$, all positions receive the same value irrespective of $i$. For $\text{Avg}_{\leftarrow}$, we output the average over all previous positions.

### 5.2  First

We can construct an attention layer using positional encoding $(-1)^i$ and an FFN layer to implement a first predicate, which is 1 at the first position and 0 everywhere else. This allows us to distinguish the first position and to compute the value $1/i$ in every position while only using the positional encoding $(-1)^i$, which are useful for technical constructions Yang et al. (2026).

$$\text{first} : \mathbb{R}^+ \xrightarrow{\text{lp}} \mathbb{R}^+$$

$$\text{first}\big((-1)^1, (-1)^2, \ldots, (-1)^n\big) = (1, 0, \ldots, 0).$$

To mark the first position, we just need to use uniform future-masked attention.

$$\text{first.}\mathbf{W}^{(Q)} = \mathbf{0} \qquad \text{first.}\mathbf{W}^{(K)} = \mathbf{0} \qquad \text{first.}\mathbf{W}^{(V)} = -\mathbf{I}$$

If $i = 1$, then only the first position receives attention, and then the output value $c_i$ is 1. If $i > 1$, then the attention output $c_i$ is the average of $-1$'s and 1's, and so will always be at most $1/3$. Finally, we can map values in $[0, 1/3]$ to 0 via a comparison operation $\text{GTZero}_{1/3}(c_i - 1/3)$, slightly modifying the construction for $\text{GTZero}$ by modifying the bias term and output sign (Section 4.7).

### 5.3  Index Lookup

Theoretical constructions of transformers often require a query to focus on a single position in order to retrieve data from that position. In this section, we will discuss how one can perform this retrieval from specific positions via the attention mechanism.

Assume that at each position $i \in [n]$, we have:

| Approach | Map | Width | Requirements |
|----------|-----|-------|--------------|
| One-hot | $(\mathbb{R}^{2N+1})^+ \overset{\text{lp}}{\to} \mathbb{R}^+$ | $\Theta(N)$ | one-hot positional encodings |
| Almost-orthogonal | $(\mathbb{R}^{2m+1})^+ \overset{\text{lp}}{\to} \mathbb{R}^+$ | $\Theta(\log N)$ | near-orthogonal positional vectors |
| Layernorm-hash | $(\mathbb{R}^9)^+ \overset{\text{lp}}{\to} \mathbb{R}^+$ | $\Theta(1)$ | Selective layernorm |
| Quadratic maximization | $(\mathbb{R}^5)^+ \overset{\text{lp}}{\to} \mathbb{R}^+$ | $\Theta(1)$ | positional features $j$ and $j^2$ where $j \in \mathbb{N}$ |

Table 5: Summary of attention-based index-lookup implementations.

- a query $q_i$, which is (an encoding of) a position, that is, $q_i \in [n]$,

- (an encoding of) $i$ itself, and

- a value $v_i$.

An *index lookup* is a self-attention layer in which each position $i$ attends to position $q_i$, making it possible to retrieve $v_{q_i}$ at each position $i$.

The following implementations of table lookup have a form similar to:

$$\text{lookup}\colon (\mathbb{R}^3)^+ \overset{\text{lp}}{\to} \mathbb{R}^+$$

$$\text{lookup}\left(\begin{bmatrix} q_1 \\ 1 \\ v_1 \end{bmatrix}, \ldots, \begin{bmatrix} q_n \\ n \\ v_n \end{bmatrix}\right) = (v_{q_1}, \ldots, v_{q_n}).$$

These implementations use average-hard attention; however, in Section 5.7 we will discuss ways to make these implementations work with soft attention as well.

### 5.3.1 One-Hot Encodings

A naïve approach to maximize attention weights at a desired position is to encode positions as one-hot vectors $\mathbf{e}_1, \ldots, \mathbf{e}_N \in \mathbb{R}^N$, up to a maximum length $N$.

$$\text{ohLookup}\colon (\mathbb{R}^{2N+1})^+ \overset{\text{lp}}{\to} \mathbb{R}^+$$

$$\text{ohLookup}\left(\begin{bmatrix} \mathbf{e}_{q_1} \\ \mathbf{e}_1 \\ v_1 \end{bmatrix}, \ldots, \begin{bmatrix} \mathbf{e}_{q_n} \\ \mathbf{e}_n \\ v_n \end{bmatrix}\right) = (v_{q_1}, \ldots, v_{q_n}).$$

The query, key, and value projections are set to

$$\mathbf{W}^{(\text{Q})} = \begin{bmatrix} \mathbf{I}^{N \times N} & \mathbf{0}^{N \times N} & \mathbf{0}^{N \times 1} \end{bmatrix}$$

$$\mathbf{W}^{(\text{K})} = \begin{bmatrix} \mathbf{0}^{N \times N} & \mathbf{I}^{N \times N} & \mathbf{0}^{N \times 1} \end{bmatrix}$$

$$\mathbf{W}^{(\text{V})} = \begin{bmatrix} \mathbf{0}^{1 \times N} & \mathbf{0}^{1 \times N} & 1 \end{bmatrix}$$

so that, at any position $i$,

$$\mathbf{q}_i = \mathbf{W}^{(\text{Q})}\left(\begin{bmatrix} \mathbf{e}_{q_i} \\ \mathbf{e}_i \\ v_i \end{bmatrix}\right) = \mathbf{e}_{q_i} \qquad \mathbf{k}_j = \mathbf{W}^{(\text{K})}\left(\begin{bmatrix} \mathbf{e}_{q_j} \\ \mathbf{e}_j \\ v_j \end{bmatrix}\right) = \mathbf{e}_j \qquad \mathbf{v}_j = \mathbf{W}^{(\text{V})}\left(\begin{bmatrix} \mathbf{e}_{q_j} \\ \mathbf{e}_j \\ v_j \end{bmatrix}\right) = v_j. \tag{36}$$

Then the dot products satisfy $\mathbf{q}_i \cdot \mathbf{k}_j = \mathbf{e}_{q_i} \cdot \mathbf{e}_j = \mathbb{I}[j = q_i]$. With hard attention, this suffices to concentrate all attention on $j = q_i$. This realizes the lookup task with width $\Omega(N)$.

With soft attention, the lookup will be only approximate, but note that there is a minimum gap between the highest attention score and the next-highest attention score of $\gamma = 1$. In Section 5.7, we will discuss how to magnify this gap enough to correct the approximation error.

### 5.3.2 Almost Orthogonal Embeddings

The one-hot approach forces the embedding width to be at least $\Omega(N)$. To mitigate this, we can use *almost orthogonal* vectors (Bhattamishra et al., 2024; Sanford et al., 2023; 2024a) to obtain width $O(\log N)$. A family of vectors $\mathbf{x}_1, \ldots, \mathbf{x}_N \in \mathbb{R}^m$ is almost orthogonal if, for some small $\epsilon > 0$,

$$|\mathbf{x}_i \cdot \mathbf{x}_j| \leq \epsilon \quad (i \neq j) \qquad\qquad \mathbf{x}_i \cdot \mathbf{x}_i \geq 1 - \epsilon. \tag{37}$$

One can pack exponentially many, $\exp(\Omega(m))$, such vectors (Vershynin, 2018, Chap. 3) into $m$ dimensions. A straightforward way to construct such vectors is: Given

- a maximum error $\epsilon > 0$,
- a maximum length $N$, and
- a maximum probability of failure $\delta > 0$,

set constant $k > 0$ such that the probability of failure is $\frac{1}{N^k} \leq \delta$ and, take positive integer $m$ as,

$$m = \left\lceil \frac{12k}{\epsilon^2} \log(2N) \right\rceil = O\left(\log N\right). \tag{38}$$

Then, if one samples $\mathbf{x}_1, \ldots, \mathbf{x}_N \in \{\pm 1/\sqrt{m}\}^m$ uniformly and independently, then with probability at most $1 - \delta$, these vectors will satisfy Eq. (37).

The construction of these almost orthogonal vectors is also equivalent to taking the Johnson–Lindenstrauss (JL) transformations (Johnson et al., 1984) of the $N$ one-hot vectors. To avoid storing $\Theta(N)$ vectors explicitly, one can use a derandomization of the JL lemma (Sivakumar, 2002) to generate them in log-space. Such near-orthogonal families have been used in constructions for sparse averaging (Sanford et al., 2023), string equality and nearest-neighbour algorithms (Bhattamishra et al., 2024), and graph algorithms (Sanford et al., 2024a).

Then we can define

$$\mathsf{aoLookup}\colon (\mathbb{R}^{2m+1})^+ \xrightarrow{\mathrm{lp}} \mathbb{R}^+$$

$$\mathsf{aoLookup}\left( \begin{bmatrix} \mathbf{x}_{q_1} \\ \mathbf{x}_1 \\ v_1 \end{bmatrix}, \ldots, \begin{bmatrix} \mathbf{x}_{q_n} \\ \mathbf{x}_n \\ v_n \end{bmatrix} \right) = (v_{q_1}, \ldots, v_{q_n}).$$

where the family $\{\mathbf{x}_1, \ldots, \mathbf{x}_N\}$ satisfies (37) with $m = O(\log N)$.

The query, key, and value projections are as in Eq. (36), so that for any position $i$, by (37), the attention dot products satisfy

$$\mathbf{q}_i \cdot \mathbf{k}_j = \mathbf{x}_{q_i} \cdot \mathbf{x}_j \begin{cases} \geq 1 - \epsilon & \text{if } j = q_i, \\ \leq \epsilon & \text{if } j \neq q_i. \end{cases}$$

Taking $\epsilon = 1/4$ for simplicity, we have $\mathbf{q}_i \cdot \mathbf{k}_j \geq 3/4$ for $j = q_i$ and $\mathbf{q}_i \cdot \mathbf{k}_j \leq 1/4$ for $j \neq q_i$. Under hard attention, the output at position $i$ is then exactly $v_{q_i}$.

With soft attention, the attention weights are nonzero at all positions, so the retrieved value will only be approximate. The minimum gap between the attention scores $\mathbf{q}_i \cdot \mathbf{k}_{q_i}$ and $\mathbf{q}_i \cdot \mathbf{k}_j$ for $j \neq q_i$ is $\gamma = 1 - 2\epsilon$. In Section 5.7, we will discuss how to magnify this gap enough to correct the approximation error.

### 5.3.3 Layernorm hash

Another way to implement table lookup is to encode positions by their *layer-norm hash* (Merrill & Sabharwal, 2024a; Yao et al., 2021). Merrill & Sabharwal (2024a) use table lookup to simulate a Turing machine tape with chain-of-thought transformers: specifically, to retrieve the last value written to a previous index on

the tape. Merrill & Sabharwal (2024b) use it to implement a binary tree construction with log-depth transformers.

$$\mathsf{lhLookup}\colon (\mathbb{R}^9)^+ \overset{\mathrm{lp}}{\to} \mathbb{R}^+$$

$$\mathsf{lhLookup}\left( \begin{bmatrix} \mathrm{lh}(q_1) \\ \mathrm{lh}(1) \\ v_1 \end{bmatrix}, \ldots, \begin{bmatrix} \mathrm{lh}(q_n) \\ \mathrm{lh}(n) \\ v_n \end{bmatrix} \right) = (v_{q_1}, \ldots, v_{q_n}).$$

$$\mathsf{lhLookup}.\mathbf{W}^{(\mathrm{Q})} = \begin{bmatrix} \mathbf{I}^{4\times 4} & \mathbf{0}^{4\times 4} & \mathbf{0}^{4\times 1} \end{bmatrix}$$

$$\mathsf{lhLookup}.\mathbf{W}^{(\mathrm{K})} = \begin{bmatrix} \mathbf{0}^{4\times 4} & \mathbf{I}^{4\times 4} & \mathbf{0}^{4\times 1} \end{bmatrix}$$

$$\mathsf{lhLookup}.\mathbf{W}^{(\mathrm{V})} = \begin{bmatrix} \mathbf{0}^{1\times 4} & \mathbf{0}^{1\times 4} & 1 \end{bmatrix}.$$

Let $\mathsf{LN}$ be a layer normalization with $\mathsf{LN}.\epsilon = 0$, $\mathsf{LN}.\beta = 0$, $\mathsf{LN}.\gamma = 1$. Merrill & Sabharwal (2024a)[2] store an integer $x$ as

$$\mathrm{lh}(x) = \mathsf{LN}\left( \begin{bmatrix} x \\ 1 \\ -x \\ -1 \end{bmatrix} \right) = \sqrt{\frac{2}{x^2+1}} \begin{bmatrix} x \\ 1 \\ -x \\ -1 \end{bmatrix}.$$

The layernorm hash is scale-invariant in the sense that $\mathrm{lh}(kx) = \mathrm{lh}(x)$. So even if we're only able to compute $x/i$ and $1/i$ (as is common when counting positions using uniform attention), we can still compute $\mathrm{lh}(x)$ as

$$\mathrm{lh}(x) = \mathsf{LN}\left( \begin{bmatrix} x/i \\ 1/i \\ -x/i \\ -1/i \end{bmatrix} \right).$$

If, for all $i > 0$, we can compute queries and keys

$$\mathbf{q}_i = \mathrm{lh}(q_i) = \mathsf{LN}\left( \begin{bmatrix} q_i/i \\ 1/i, -q_i/i \\ -1/i \end{bmatrix} \right), \qquad \mathbf{k}_j = \mathrm{lh}(j) = \mathsf{LN}\left( \begin{bmatrix} 1 \\ 1/j \\ -1 \\ -1/j \end{bmatrix} \right)$$

then the dot product $s_{i,j} = \mathbf{q}_i \cdot \mathbf{k}_j = \mathrm{lh}(q_i) \cdot \mathrm{lh}(j)$ is uniquely maximized when $q_i = j$. Under hard attention, this allows us to attend only to position $j = q_i$, after which the the value of $v_j$ can be retrieved by appropriately setting $\mathbf{W}^{(\mathrm{V})}$. But because the minimum gap between the score at the desired position $s_{i,q_i}$ and the score at other positions decreases with $n$, a considerable amount of error accumulates when approximating ths construction using soft attention Section 5.7.

One problem is that the residual stream may store other values besides the ones shown above, but standard pre-norm is applied to all values in the residual stream. Then lh will incorrectly be scaled by some factor that may be different at each position. So we need the ability to selectively apply $\mathsf{LN}$ to just four components of a vector. We can do this if we use pre-norm *and* modify the architecture by inserting a linear transformation $\mathbf{W}^{(\mathrm{N})}$ before the layer normalization:

$$\mathbf{y} = \mathsf{sa}(\mathsf{LN}(\mathbf{W}^{(\mathrm{N})}\mathbf{x})) + \mathbf{x}.$$

The linear transformation $\mathbf{W}^{(\mathrm{N})}$, which can be restricted to a diagonal matrix if desired, can mask out information encoded in the residual stream that is not relevant at this layer, allowing the network to compute the layer-norm hash of a specific value.

---

[2]Merrill & Sabharwal (2024a) use a slightly different formula for $\mathsf{LN}$, but this only changes the construction by a factor of 2.

### 5.3.4 Quadratic Maximization

Barcelo et al. (2024) include $j$ and $j^2$ in the position embedding, allowing table lookup as follows:

$$\mathsf{qmLookup}\colon (\mathbb{R}^5)^+ \overset{\mathrm{lp}}{\to} \mathbb{R}^+$$

$$\mathsf{qmLookup}\left(\begin{bmatrix} q_1 \\ 1 \\ 1 \\ 1^2 \\ v_1 \end{bmatrix}, \begin{bmatrix} q_2 \\ 1 \\ 2 \\ 2^2 \\ v_2 \end{bmatrix}, \ldots, \begin{bmatrix} q_n \\ 1 \\ n \\ n^2 \\ v_n \end{bmatrix}\right) = (v_{q_1}, v_{q_2}, \ldots, v_{q_n}).$$

$$\mathsf{qmLookup}.\mathbf{W}^{(\mathrm{Q})} = \begin{bmatrix} 1 & 0 & 0 & 0 & 0 \\ 0 & 1 & 0 & 0 & 0 \end{bmatrix}$$

$$\mathsf{qmLookup}.\mathbf{W}^{(\mathrm{K})} = \begin{bmatrix} 0 & 0 & 2 & 0 & 0 \\ 0 & 0 & 0 & -1 & 0 \end{bmatrix}$$

$$\mathsf{qmLookup}.\mathbf{W}^{(\mathrm{V})} = \begin{bmatrix} \mathbf{0}^{1\times 4} & 1 \end{bmatrix}.$$

Then the queries and keys are

$$\mathbf{q}_i = \begin{bmatrix} q_i \\ 1 \end{bmatrix}, \qquad \mathbf{k}_j = \begin{bmatrix} 2j \\ -j^2 \end{bmatrix}$$

Their dot product $s_{i,j}$ is $2q_i j - j^2$, which is uniquely maximized when $j = q_i$. This is true even if either $q_i$ or $k_j$ is scaled by some factor (for example, $1/i$ or $1/n$). Under hard attention, this solves the lookup problem exactly.

Under soft attention, the lookup is only approximate. But the minimum gap between the score at the desired position $s_{i,q_i}$ and the score at any other position $s_{i,j}$ ($j \neq q_i$) is $\gamma = 1$, and we will discuss in Section 5.7 how to magnify this gap enough to correct the approximation error.

### 5.4 Predecessor

The predecessor function is the special case of index lookup where each position $i$ attends to position $i-1$, and the first position always outputs 0. The construction could be generalized to output a default value other than 0, or even output the original value in the first position. The index lookup methods of Section 5.3 can be used to do this; in particular, using quadratic maximization (Section 5.3.4), $\mathbf{W}^{(\mathrm{K})}$ and $\mathbf{W}^{(\mathrm{Q})}$ can be set so that $s_{i,j}$ is $2(i-1)j - j^2$. Then, the first position can be set to 0 using the construction from Section 5.2 if the positional encoding includes $(-1)^i$.

In this section, we present two alternative constructions. First, we present one that uses a simpler position encoding but potentially requiring more layers. Here, we assume that the values $v_i$ are bounded to $[0, 1]$. Then, we present a simpler construction which takes advantage of strict future-masking and does not require values to be in $[0, 1]$.

$$\mathsf{pred}\colon (\mathbb{R}^2 \times [0,1])^+ \to [0,1]^+$$

$$\mathsf{pred}\left(\begin{bmatrix} 1 \\ (-1)^1 \\ v_1 \end{bmatrix}, \begin{bmatrix} 1 \\ (-1)^2 \\ v_2 \end{bmatrix}, \ldots, \begin{bmatrix} 1 \\ (-1)^n \\ v_n \end{bmatrix}\right) = (0, v_1, \ldots, v_{n-1}).$$

This can be achieved by rightmost UHAT with positional encoding $(-1)^i$. We compute the predecessor at positions $i > 1$ by making each position $i$ attend only to position $(i-1)$. If $i = 1$, we output 0. If $i$ is odd (and greater than 1), we make the attention scores greater at even positions, so that rightmost tie-breaking selects position $(i-1)$. If $i$ is even, we make the attention scores greater at odd positions.

To do this, we use two future-masked attention layers (or one layer with two heads). The first is:

$$\mathsf{pred.odd}.\mathbf{W}^{(\mathrm{Q})} = \begin{bmatrix} 1 & 0 & 0 \end{bmatrix} \qquad \mathsf{pred.odd}.\mathbf{W}^{(\mathrm{K})} = \begin{bmatrix} 0 & 1 & 0 \end{bmatrix} \qquad \mathsf{pred.odd}.\mathbf{W}^{(\mathrm{V})} = \begin{bmatrix} 0 & 0 & 1 \end{bmatrix}$$

If $i$ is odd, the result of $\mathsf{pred.odd}$ is $v_{i-1}$. The second attention head is defined similarly:

$$\mathsf{pred.even}.\mathbf{W}^{(\mathrm{Q})} = \begin{bmatrix} 1 & 0 & 0 \end{bmatrix} \qquad \mathsf{pred.even}.\mathbf{W}^{(\mathrm{K})} = \begin{bmatrix} 0 & -1 & 0 \end{bmatrix} \qquad \mathsf{pred.even}.\mathbf{W}^{(\mathrm{V})} = \begin{bmatrix} 0 & 0 & 1 \end{bmatrix}$$

Now if $i$ is even, the result of $\mathsf{pred.even}$ is $v_{i-1}$.

At each position $i$, we can check if $i = 1$ using the construction in Section 5.2, and we can check if $i$ is even using $\mathsf{GTZero}_1((-1)^i)$. After that, because the $v_j$ are bounded, we can use a conditional (Section 4.9) in order to select the correct result either from $\mathsf{pred.even}$ or $\mathsf{pred.odd}$:

$$\mathsf{pred}(\mathbf{z}_1, \ldots \mathbf{z}_n) = \mathsf{if}(\mathsf{GTZero}_1((-1)^i), \mathsf{pred.even}(\mathbf{z}_1, \ldots \mathbf{z}_n), \mathsf{pred.odd}(\mathbf{z}_1, \ldots \mathbf{z}_n)).$$

This construction is similar to the one in Barcelo et al. (2024) using $\mathsf{AHAT}$, and we note that Yang et al. (2026) demonstrate how this can be simulated using $\mathsf{SMAT}$, if the values are bounded.

If we choose to use strict future-masking, then predecessor look-up becomes considerably easier. This is because for any position $i$, the attention weight will only be non-negative at positions $j < i$. In particular, it suffices to use a positional encoding of $i/n$ along with $\mathsf{AHAT}$. Then, letting the attention matrices act as linear projections on the $i/n$ positional encoding, the $i - 1$ position will automatically be selected by $\mathsf{AHAT}$ as it will be the largest among the sequence $\mathsf{AHAT}(1/n, \ldots, (i-1)/n, 0, \ldots, 0)$, where note that the value $i/n$ is not present because the attention is strictly future-masking. If we use rightmost tie-breaking and $\mathsf{UHAT}$, the construction is even simpler; set the attention matrices to 0, and since the attention weights will all be equal, rightmost tie-breaking (with strict future-masking) will directly select the predecessor position.

## 5.5   Simulating Multi-Head Attention with Single-Head Attention

The result of a multi-headed attention layer with $H$ heads with $d_{\mathrm{hid}}$ key/value dimensions per head is the concatenation of resulting vectors from $H$ separate attention heads. In this way, an attention layer with $H$ heads can be simulated by $H$ single-headed attention layers, and the results summed together at the end using Section 4.5.

## 5.6   Simulating Unique-Hard Attention with Average-Hard Attention

When using *average-hard* attention, sometimes we want to be able to simulate *leftmost unique-hard* or *rightmost unique-hard* attention. To do this, we have to add a *tie-breaking* term to the attention scores that decreases or increases (respectively) with the position. However, care must be taken not to make the tie-breaking term so big that a non-maximal attention score becomes the maximal attention score.

Suppose that we have attention scores computed from queries and keys as follows:

$$s_{ij} = \frac{\mathbf{q}_i \cdot \mathbf{k}_j}{\sqrt{d_{\mathrm{key}}}}$$

and we know that there exists $\gamma > 0$ such that for all $i, j$, either

$$s_{ij} = \max_{j'} s_{ij'} \quad \text{or} \quad s_{ij} < \max_{j'} s_{ij'} - \gamma. \tag{39}$$

First, depending on the desired form of unique-hard attention, the appropriate tie-breaking term $t$ can be chosen:

$$t(j) = \begin{cases} -\dfrac{1}{j}, & \text{rightmost} \\[2mm] \dfrac{1}{j}, & \text{leftmost} \\[2mm] \dfrac{j}{n}, & \text{rightmost} \\[2mm] -\dfrac{j}{n}, & \text{leftmost} \end{cases}$$

Second, the appropriate $\gamma$ must be chosen to satisfy Eq. (39). Then, we add additional dimensions to the input embedding so that $\gamma$ and $t(i)$ can be stored in each position. Thus, an input of $\begin{bmatrix} \mathbf{x} \end{bmatrix}$ at position $i$ becomes $\begin{bmatrix} \mathbf{x} \\ \gamma \\ t(i) \end{bmatrix}$. Then, the tie breaking term is added in to the scores $\hat{s_{ij}} = s_{ij} + \gamma t(j)$ by setting the key, query, and value matrices as follows:

$$\hat{\mathbf{W}}^{(\mathrm{Q})} = \begin{bmatrix} \mathbf{W}^{(\mathrm{Q})} & \mathbf{0} & \mathbf{0} \\ \mathbf{0} & 1 & 0 \end{bmatrix} \qquad \hat{\mathbf{W}}^{(\mathrm{K})} = \begin{bmatrix} \mathbf{W}^{(\mathrm{Q})} & \mathbf{0} & \mathbf{0} \\ \mathbf{0} & 0 & 1 \end{bmatrix} \qquad \hat{\mathbf{W}}^{(\mathrm{V})} = \begin{bmatrix} \mathbf{W}^{(\mathrm{V})} & \mathbf{0} & \mathbf{0} \end{bmatrix}.$$

### 5.7 Simulating Tieless Average-Hard Attention with Soft Attention

Many of the above constructions are exact under average-hard attention but only approximate under soft attention. In particular, to perform index-lookup operations, attention has to be concentrated on the single query position. With hard attention, 100% of the score can be concentrated on the query position, with soft attention this is not possible, because every position receives positive attention. This causes error when approximating hard attention using soft attention.

The amount of error is based on the gap $\gamma$ between the highest attention score and all other attention scores. We have noted, when appropriate, what the minimum gap $\gamma$ will be in many constructions. In this section, we discuss how to correct this error if $\gamma$ is known.

One way to correct the approximation error is using limited-precision arithmetic. If we multiply the query vectors by a factor $1/\tau$ large enough that $\exp{-\gamma/\tau}$ rounds to zero, then softmax becomes exactly equal to ahardmax and soft attention becomes equivalent to average-hard attention.

When we want to use unbounded precision, we will also need the factor $1/\tau$ to depend on the sequence length. This results in a *non-uniform* construction (see Section 2.3). In index-lookup operations the score is always highest on a single position, a condition which we give a special name to.

**Definition 5.2.** For any vector $\mathbf{s} \in \mathbb{R}^+$, let $M = \max_i s_i$. We say that $\mathbf{s}$ is *tieless* if $|\{i \mid s_i = M\}| = 1$, and $\mathbf{s}$ has *gap* $\gamma$ if for all $i$ such that $s_i \neq M$, we have $s_i \leq M - \gamma$.

If $\mathbf{s}$ is tieless, then $\mathrm{lhardmax}(\mathbf{s}) = \mathrm{rhardmax}(\mathbf{s}) = \mathrm{ahardmax}(\mathbf{s})$, and we write $\mathrm{hardmax}(\mathbf{s})$ for all three. In the rest of this section, we assume $\mathbf{s}$ will be tieless, as it would be for the typical index-lookup setup Section 5.3.

**Lemma 5.3** (Edelman et al. 2022, Lemma B.7)**.** *Let $\mathbf{s} = (s_1, \ldots, s_n)$ be attention scores and let $j^* \in [n]$ and $\gamma > 0$ be such that for all $j \neq j^*$ we have $s_j < s_{j^*} - \gamma$. Then*

$$\| \mathrm{hardmax}(\mathbf{s}) - \mathrm{softmax}(\mathbf{s}) \|_1 \leq 2ne^{-\gamma}.$$

When we do not want dependence on the sequence length, we can assume that $v_i \in \{0, 1\}$ for all $i \in [n]$, and that there is a maximum length $N$. The idea is to keep the error small enough that we can round the retrieved valued correctly to either 0 or 1.

Let $\mathbf{W}^{(Q)'} = (1/\tau)\mathbf{W}^{(Q)}$, where $\tau = \gamma/(\log 8N)$. If we compute attention scores using $\mathbf{W}^{(Q)'}$ instead of $\mathbf{W}^{(Q)}$, we get $\mathbf{q}'_i \cdot \mathbf{k}_j = (1/\tau)\mathbf{q}_i \cdot \mathbf{k}_j$, and the minimum gap is now $\log 8N$ instead of $\gamma$. Let $\alpha_1, \ldots, \alpha_N \in [0,1]$ be the resulting attention weights at position $i$. The output of attention at position $i$ is

$$\mathbf{c}_i = \sum_{j=1}^{N} \alpha_j \mathbf{v}_j = \alpha_{q_i} v_{q_i} + \sum_{j \neq q_i} \alpha_j v_j$$

$$|\mathbf{c}_i - v_{q_i}| \leq (1 - \alpha_{q_i}) + \sum_{j \neq q_i} \alpha_j$$

$$\leq 2Ne^{-\gamma/\tau} \qquad\qquad \text{(by Lemma 5.3)}$$

$$= \frac{1}{4}.$$

So $\mathbf{c}_i \leq 1/4$ if $v_{q_i} = 0$ and $\mathbf{c}_i \geq 3/4$ if $v_{q_i} = 1$ (Bhattamishra et al., 2024, Thm. 1). Thus, the value can be rounded to 0 or 1 using a 2-layer ReLU FFN, namely, $\mathsf{GTZero}_{1/2}(\mathbf{c}_i - 1/4)$.

Sometimes we can do better; in particular, Yang et al. (2026) show how this can be done with quadratic maximization without assuming a maximum length $N$.

## 6 Layer Normalization

Layer normalization (Ba et al., 2016), or layernorm for short, is a normalization technique originally proposed for the purpose of reducing training time.

**Definition 6.1** (layer normalization, Ba et al., 2016)**.** A *layer normalization* is a function

$$\mathsf{LN} \colon \mathbb{R}^d \to \mathbb{R}^d$$

$$\mathsf{LN}(x) = \frac{x_i - \mu}{\sqrt{\sigma^2 + \mathsf{LN}.\varepsilon}} \odot \mathsf{LN}.\gamma + \mathsf{LN}.\beta$$

$$\mu = \frac{1}{d} \sum_{i=1}^{d} x_i$$

$$\sigma^2 = \frac{1}{d} \sum_{i=1}^{d} (x_i - \mu)^2$$

where $\mathsf{LN}.\gamma, \mathsf{LN}.\beta \in \mathbb{R}^d$ and $\mathsf{LN}.\varepsilon > 0$ is a small constant for numerical stability.

The original transformer model (Vaswani et al., 2017) used layer normalization (or layernorm for short) after each sublayer and residual connection. It has since been shown to have an impact on the expressiveness of the model, for example, by affecting the Lipschitz continuity of the model (Hahn, 2020) and the ability to express certain attention patterns (Brody et al., 2023). In theoretical work, layernorm in some cases may complicate the proof, and in other cases may be an essential component of the proof. In this section, we discuss the theoretical aspects of layer normalization and how the literature on expressivity proofs has treated it.

### 6.1 Relevant Properties

The use of layer normalization can affect the sensitivity of a network with respect to its inputs. Understanding how sensitive a network is to small perturbations in its input is crucial for analyzing its stability and generalization. The property of Lipschitz continuity provides a formal way to bound this sensitivity, and is useful in analysis in adversarial robustness (Zühlke & Kudenko, 2025), generalization (Bartlett et al., 2017), and expressivity (Chiang et al., 2023).

**Definition 6.2** (*k*-Lipschitz Continuity)**.** A function $f$ is *k-Lipschitz continuous* if for all $\mathbf{x}_1, \mathbf{x}_2$, $\|f(\mathbf{x}_1) - f(\mathbf{x}_2)\| \leq k\|\mathbf{x}_1 - \mathbf{x}_2\|$.

In simple terms, a $k$-Lipschitz continuous function cannot change arbitrarily fast. Since every operation in an FFN (matrix multiplication, addition, and the ReLU activation function) is itself Lipschitz continuous, their composition is as well. We discuss below how layernorm may break the Lipschitz continuity of a network.

Another important property, particularly for FFNs without bias terms, is how the function's output scales with the magnitude of its input. This scaling behavior is captured by the concept of positive homogeneity.

**Definition 6.3** (Positive $k$-homogeneity)**.** A function $f$ is *positively $k$-homogeneous* for all $\mathbf{x}$ and $c \geq 0$, $f(c\mathbf{x}) = c^k f(\mathbf{x})$.

This definition states that scaling the input vector by a non-negative constant $c$ results in the output being scaled by $c^k$. For a standard FFN with ReLU activations and no bias terms (i.e., $\mathsf{ff}.\mathbf{b}_1 = \mathsf{ff}.\mathbf{b}_2 = \mathbf{0}$), the function is positively 1-homogeneous (or "linear" in its scaling behavior). This is because $\mathrm{ReLU}(cx) = c \cdot \mathrm{ReLU}(x)$ for all $c \geq 0$, and this property is preserved through the linear transformations of the network. If every layer of a network is 1-homogeneous and the output is scale-invariant, then layernorm will not affect the network.

## 6.2 Post-Norm vs. Pre-Norm

The original definition of the transformer used what is now known as a "post-norm" architecture, where layer normalization is applied after each residual connection:

$$\mathbf{y} = \mathsf{LN}(f(\mathbf{x}) + \mathbf{x})$$

where $f$ is either a self-attention or FFN.

This is in contrast to the "pre-norm" architecture, where layer normalization is applied before each residual connection:

$$\mathbf{y} = f(\mathsf{LN}(\mathbf{x})) + \mathbf{x}.$$

Additionally, the final layer is followed by one more layer normalization.

The pre-norm architecture is currently the standard. This is because the post-norm architecture has problems with exploding gradients, making training unstable, which the pre-norm architecture does not (Xiong et al., 2020). Some constructions, however, still use post-norm (Chiang & Cholak, 2022; Chiang et al., 2023; Hahn & Rofin, 2024; Yang & Chiang, 2024).

## 6.3 Circumventing Layer Normalization

Some theoretical constructions simply omit layernorm, and some constructions don't do anything useful with it, but simply try to circumvent it. We can't circumvent it completely; the best we can do is the following.

For any layernorm $\mathsf{LN}$ with $\mathsf{LN}.\beta = \mathbf{0}$, for all $\mathbf{x}$,

$$\mathsf{LN}\left(\begin{bmatrix} \mathbf{x} \\ -\mathbf{x} \end{bmatrix}\right) = \begin{bmatrix} c\mathbf{x} \\ -c\mathbf{x} \end{bmatrix}$$

for some $c$. That is, if the components of a vector come in pairs that are additive inverses of each other, then it has zero mean, so the layernorm $\mathsf{LN}$ can only scale the vector.

## 6.4 Sign Function

Hahn (2020) showed that if a transformer's position-wise operations are all Lipschitz-continuous and its position embeddings are bounded, then it also has the Lipschitz-like property that a change in a single input symbol can only produce a change of $O(1/n)$ in any output activation.

But layer normalization $\mathsf{LN}$ is Lipschitz-continuous only if $\mathsf{LN}.\epsilon > 0$; if (as originally defined) $\mathsf{LN}.\epsilon = 0$, then $\mathsf{LN}$ is not Lipschitz-continuous, possibly allowing the model to express more complex functions. For example, Yang & Chiang (2024) use layernorm with $\mathsf{LN}.\epsilon = 0$ in order to compare numbers (like in Section 4.7) via computation of a sign function.

Here, we explain how layernrom can be used to compute the sign function (here, the computation outputs 0 on inputs of 0). The primary challenge is that layernorm cannot be used only on selected components. In order to use layernorm to compute the sign function, we have to clip *all* components to be in $\{-\delta, \delta\}$ and ensure the entire vector has mean 0. Let $\mathbf{x} \in \mathbb{R}^d$ be a vector with $|x_c| \geq \delta$ for all $c \in [d]$. First, we ensure the vector has 0 mean by doubling up the dimensions and negating every other one. A vector $\mathbf{x}_1, \mathbf{x}_2, \ldots, \mathbf{x}_n$ thus becomes $-\mathbf{x}_1, \mathbf{x}_1, -\mathbf{x}_2, \mathbf{x}_2, \ldots, -\mathbf{x}_n, \mathbf{x}_n$. We can construct a FFN that clips all activations to $\pm\delta$ using the following weights and biases:

$$\mathsf{clip}_\delta \colon \mathbb{R} \to \mathbb{R}$$

$$\mathsf{clip}_\delta.\mathbf{W}_1 = \begin{bmatrix} 1 \\ 1 \end{bmatrix} \qquad\qquad \mathsf{clip}_\delta.\mathbf{b}_1 = \begin{bmatrix} \delta \\ -\delta \end{bmatrix}$$

$$\mathsf{clip}_\delta.\mathbf{W}_2 = \begin{bmatrix} 1 & -1 \end{bmatrix} \qquad\qquad \mathsf{clip}_\delta.\mathbf{b}_2 = \begin{bmatrix} -\delta \end{bmatrix}.$$

Applying layernorm after this FFN will normalize all activations to $\pm 1$.

$$\mathsf{LN} \colon \mathbb{R} \to \mathbb{R}$$
$$\mathsf{LN}.\epsilon = 0$$
$$\mathsf{LN}.\beta = 0$$
$$\mathsf{LN}.\gamma = 1.$$

# 7 Rounding and Approximation

## 7.1 Rounding

Transformers as defined in Section 2 operate over real-valued activations, which sometimes comes in tension with the discrete tasks we expect them to do, such as simulating finite automata or logical reasoning. One way to address this is to enforce rounding to fixed-precision. This may not be cause for objection due to the fact that transformers are implemented using fixed-precision in practice (say 32-bit floating point for example). In this section, we describe how rounding can be used as a mechanism, and how it may be handled via approximation.

### 7.1.1 Comparisons

In Section 4.7 we saw that a FFN can simulate a step function up to a fixed tolerance $\epsilon$.

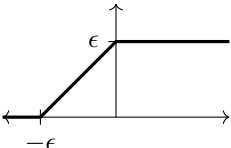

If values smaller than $\epsilon$ were rounded down to 0, this would permit exact simulation of a step function.

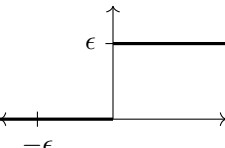

This could be scaled up to perform comparisons exactly.

## 7.2 Error Analysis

While a transformer may sometimes be unable to exactly implement a discrete operation, it can often approximate the operation very closely. When a transformer construction makes such an approximation,

we can bound the error using the following results. Below, $\|\cdot\|$ around matrices is the operator norm. For simplicity, one can use the $L_{\infty,1}$ norm instead,

$$\|\mathbf{A}\|_{\infty,1} = \sum_i \max_j |\mathbf{A}_{i,j}|.$$

**Proposition 7.1** (Bounded activations, Hahn 2020, Chiang et al. 2023). *Let* tf *be a transformer with* tf.pe$_n(i) \leq P$ *for all $n$ and $i \in [n]$. Even if* tf *contains a layer normalization* LN *with* LN.$\epsilon = 0$, *there is an $X$ such that for all $\ell$ and $i$, we have $\|\mathbf{z}_i^{(\ell)}\| \leq X$.*

**Proposition 7.2** (Error analysis of FFNs). *If $\|\hat{\mathbf{x}} - \mathbf{x}\| \leq \delta$ and* ff *is a ReLU FFN, then there is a constant $K$ such that*

$$\|\text{ff}(\hat{\mathbf{x}}) - \text{ff}(\mathbf{x})\| \leq K\delta.$$

**Proposition 7.3** (Error analysis of attention, Yang et al. 2026). *If $\|\mathbf{x}\| \leq X$, $\|\hat{\mathbf{x}} - \mathbf{x}\| \leq \delta \leq 1$ and* sa *is a self-attention layer, then there is a constant $K$ such that for all $i$,*

$$\|[\text{sa}(\hat{\mathbf{x}})]_i - [\text{sa}(\mathbf{x})]_i\| \leq K\delta.$$

**Proposition 7.4** (Error analysis of residual connections). *If $\|\hat{\mathbf{x}} - \mathbf{x}\| \leq \delta$, $f \colon \mathbb{R}^d \to \mathbb{R}^d$, and $\|f(\hat{\mathbf{x}}) - f(\mathbf{x})\| \leq \epsilon$, then*

$$\|f(\hat{\mathbf{x}}) + \hat{\mathbf{x}} - (f(\mathbf{x}) + \mathbf{x})\| \leq \epsilon + \delta.$$

**Proposition 7.5** (Error analysis of layer normalization). *If $\|\mathbf{x}\| \leq X$, $\|\hat{\mathbf{x}} - \mathbf{x}\| \leq \delta \leq 1$, and* LN *is a layer normalization with* LN.$\epsilon > 0$, *then there is a constant $K$ such that*

$$\|\text{LN}(\hat{\mathbf{x}}) - \text{LN}(\mathbf{x})\| \leq K\delta.$$

# 8 Assembly

## 8.1 Residual stream

The layers of a transformer tf compute a sequence of sequences of vectors,

$$
\begin{array}{rcll}
(\bar{\mathbf{c}}_1^{(1)}, \ldots, \bar{\mathbf{c}}_n^{(1)}) & = & \text{tf.tl}^{(1)}.\text{sa}(\mathbf{z}_1^{(0)}, \ldots, \mathbf{z}_n^{(0)}) & + \quad (\mathbf{z}_1^{(0)}, \ldots, \mathbf{z}_n^{(0)}) \\
(\mathbf{z}_1^{(1)}, \ldots, \mathbf{z}_n^{(1)}) & = & \text{tf.tl}^{(1)}.\text{ff}(\bar{\mathbf{c}}_1^{(1)}, \ldots, \bar{\mathbf{c}}_n^{(1)}) & + \quad (\bar{\mathbf{c}}_1^{(1)}, \ldots, \bar{\mathbf{c}}_n^{(1)}) \\
& \vdots & & \\
(\bar{\mathbf{c}}_1^{(L)}, \ldots, \bar{\mathbf{c}}_n^{(L)}) & = & \text{tf.tl}^{(L)}.\text{sa}(\mathbf{z}_1^{(L-1)}, \ldots, \mathbf{z}_n^{(L-1)}) & + \quad (\mathbf{z}_1^{(L-1)}, \ldots, \mathbf{z}_n^{(L-1)}) \\
(\mathbf{z}_1^{(L)}, \ldots, \mathbf{z}_n^{(L)}) & = & \text{tf.tl}^{(L)}.\text{ff}(\bar{\mathbf{c}}_1^{(L)}, \ldots, \bar{\mathbf{c}}_n^{(L)}) & + \quad (\bar{\mathbf{c}}_1^{(L)}, \ldots, \bar{\mathbf{c}}_n^{(L)})
\end{array}
$$

Because each layer doesn't replace its input but adds to it, this sequence of sequences of vectors is sometimes referred to as the "residual stream" (Elhage et al., 2021).

In theoretical constructions of transformers, if there is no need to minimize the dimension $d$, it's typical for each layer only to add to components that have a zero value. Then a transformer resembles a straight-line program, where each layer sets one or more components, each component is set exactly once, and each layer can see the components set by all previous layers.

## 8.2 Routing Lemma

The following lemma is extremely useful for passing information through a transformer, and constructions very frequently make use of it without mentioning it explicitly.

**Lemma 8.1** (Routing Lemma). *If* ff$\colon \mathbb{R}^d \to \mathbb{R}^d$ *is a FFN, and* $\mathbf{L}, \mathbf{R} \colon \mathbb{R}^d \to \mathbb{R}^d$ *are linear transformations, then* $\mathbf{L} \circ$ ff *and* ff $\circ \mathbf{R}$ *are also FFNs.*

*Similarly, let* sa$\colon (\mathbb{R}^d)^+ \to (\mathbb{R}^d)^+$ *be a self-attention layer, and* $\mathbf{L}, \mathbf{R} \colon \mathbb{R}^d \to \mathbb{R}^d$ *be linear transformations. Then* $\mathbf{L}$ *and* $\mathbf{R}$ *induce positionwise mappings* $(\mathbb{R}^d)^+ \to (\mathbb{R}^d)^+$, *and* $\mathbf{L} \circ$ sa *and* sa $\circ \mathbf{R}$ *are also self-attention layers.*

This means in particular that we can reorder, duplicate, or zero out the components of the hidden vectors of a transformer at will. We will leave these transformations implicit. (In particular, our definition of FFN and self-attention requires that the input and output dimension ($d$) be equal, but we will often give definitions of FFNs and self-attentions where this is not the case, because the routing lemma can be used to make them equal.)

*Proof.* Given $\mathsf{ff}$ and matrices $\mathbf{L}$ and $\mathbf{R}$, we can construct a FFN $\mathsf{ff}' = \mathbf{L} \circ \mathsf{ff} \circ \mathbf{R}$:

$$\mathsf{ff}'.\mathbf{W}_1 = (\mathsf{ff}.\mathbf{W}_1)\mathbf{R} \qquad\qquad \mathsf{ff}'.\mathbf{b}_1 = \mathsf{ff}.\mathbf{b}_1$$
$$\mathsf{ff}'.\mathbf{W}_2 = \mathbf{L}(\mathsf{ff}.\mathbf{W}_2) \qquad\qquad \mathsf{ff}'.\mathbf{b}_2 = \mathbf{L}(\mathsf{ff}.\mathbf{b}_2).$$

Given $\mathsf{sa}$ and matrices $\mathbf{L}$ and $\mathbf{R}$, we can construct a self-attention layer $\mathsf{sa}' = \mathbf{L} \circ \mathsf{sa} \circ \mathbf{R}$:

$$\mathsf{sa}'.\mathbf{W}^{(\mathrm{Q})} = (\mathsf{sa}.\mathbf{W}^{(\mathrm{Q})})\mathbf{R}$$
$$\mathsf{sa}'.\mathbf{W}^{(\mathrm{K})} = (\mathsf{sa}.\mathbf{W}^{(\mathrm{K})})\mathbf{R}$$
$$\mathsf{sa}'.\mathbf{W}^{(\mathrm{V})} = \mathbf{L}(\mathsf{sa}.\mathbf{W}^{(\mathrm{V})})\mathbf{R}. \qquad\qquad \square$$

### 8.3 Composition operations

**Lemma 8.2** (Serial Composition). *Given two layers* $\mathsf{tl}_1, \mathsf{tl}_2 \colon \mathbb{R}^d \to \mathbb{R}^d$, *there is a layer* $\mathsf{tl}_2 \circ \mathsf{tl}_1$ *that computes* $\mathsf{tl}_2(\mathsf{tl}_1(w))$.

*Proof.* Simply stack $\mathsf{tl}_2$ on top of the layers of $\mathsf{tl}_1$. This can then be extended to sequences of multiple layers. $\qquad\square$

**Lemma 8.3** (Parallel Composition). *Given transformers without layer normalization*

$$\mathsf{tf}_1 \colon \Sigma^* \xrightarrow{\mathrm{lp}} (\mathbb{R}^{d_1})^*$$
$$\mathsf{tf}_2 \colon \Sigma^* \xrightarrow{\mathrm{lp}} (\mathbb{R}^{d_2})^*$$

*there is a transformer*

$$\mathsf{tf}_1 \oplus \mathsf{tf}_2 \colon \Sigma^* \xrightarrow{\mathrm{lp}} (\mathbb{R}^{d_1+d_2})^*$$

*such that for all strings* $w = w_1 \cdots w_n \in \Sigma^*$,

$$(\mathsf{tf}_1 \oplus \mathsf{tf}_2)(w) = \begin{bmatrix} \mathsf{tf}_1(w)_1 \\ \mathsf{tf}_2(w)_1 \end{bmatrix} \cdots \begin{bmatrix} \mathsf{tf}_1(w)_n \\ \mathsf{tf}_2(w)_n \end{bmatrix}. \qquad (40)$$

*Proof.* This is a special case of Lemma 8.1. Let $d_1$ and $d_2$ be the width of $\mathsf{tf}_1$ and $\mathsf{tf}_2$, respectively, and let $d = d_1 + d_2$. If one of $\mathsf{tf}_1$ and $\mathsf{tf}_2$ has fewer layers than the other, add trivial layers (layers that compute the identity function) until they have the same number of layers $L$.

The new transformer has embedding layer

$$(\mathsf{tf}_1 \oplus \mathsf{tf}_2).\mathsf{we}(\sigma) = \begin{bmatrix} \mathsf{tf}_1.\mathsf{we}(\sigma) \\ \mathsf{tf}_2.\mathsf{we}(\sigma) \end{bmatrix}$$
$$(\mathsf{tf}_1 \oplus \mathsf{tf}_2).\mathsf{pe}_n(i) = \begin{bmatrix} \mathsf{tf}_1.\mathsf{pe}_n(i) \\ \mathsf{tf}_2.\mathsf{pe}_n(i) \end{bmatrix}.$$

For each layer $\ell \in [L]$, let $f_1 = \mathsf{tf}_1.\mathsf{tl}_\ell$ and $f_2 = \mathsf{tf}_2.\mathsf{tl}_\ell$. Widen $f_1$ into a layer $f_1'$ with width $d$ as follows.

$$f_1'.\mathsf{sa}.\mathbf{W}^{(\mathrm{Q})} = \begin{bmatrix} f_1.\mathsf{sa}.\mathbf{W}^{(\mathrm{Q})} & \mathbf{0} \end{bmatrix}$$

$$f_1'.\mathsf{sa}.\mathbf{W}^{(\mathrm{K})} = \begin{bmatrix} f_1.\mathsf{sa}.\mathbf{W}^{(\mathrm{K})} & \mathbf{0} \end{bmatrix}$$

$$f_1'.\mathsf{sa}.\mathbf{W}^{(\mathrm{V})} = \begin{bmatrix} f_1.\mathsf{sa}.\mathbf{W}^{(\mathrm{V})} & \mathbf{0} \\ \mathbf{0} & \mathbf{0} \end{bmatrix}$$

$$f_1'.\mathsf{ff}.\mathbf{W}_1 = \begin{bmatrix} f_1.\mathsf{ff}.\mathbf{W}_1 & \mathbf{0} \\ \mathbf{0} & \mathbf{0} \end{bmatrix} \qquad f_1'.\mathsf{ff}.\mathbf{b}_1 = \begin{bmatrix} f_1.\mathsf{ff}.\mathbf{b}_1 \\ \mathbf{0} \end{bmatrix}$$

$$f_1'.\mathsf{ff}.\mathbf{W}_2 = \begin{bmatrix} f_1.\mathsf{ff}.\mathbf{W}_2 & \mathbf{0} \\ \mathbf{0} & \mathbf{0} \end{bmatrix} \qquad f_1'.\mathsf{ff}.\mathbf{b}_2 = \begin{bmatrix} f_1.\mathsf{ff}.\mathbf{b}_2 \\ \mathbf{0} \end{bmatrix}$$

Similarly, widen $f_2$ into a layer $f_2'$ with width $d$, but using the bottom half of the activation vectors:

$$f_2'.\mathsf{sa}.\mathbf{W}^{(\mathrm{Q})} = \begin{bmatrix} \mathbf{0} & f_2.\mathsf{sa}.\mathbf{W}^{(\mathrm{Q})} \end{bmatrix}$$

$$f_2'.\mathsf{sa}.\mathbf{W}^{(\mathrm{K})} = \begin{bmatrix} \mathbf{0} & f_2.\mathsf{sa}.\mathbf{W}^{(\mathrm{K})} \end{bmatrix}$$

$$f_2'.\mathsf{sa}.\mathbf{W}^{(\mathrm{V})} = \begin{bmatrix} \mathbf{0} & \mathbf{0} \\ \mathbf{0} & f_2.\mathsf{sa}.\mathbf{W}^{(\mathrm{V})} \end{bmatrix}$$

$$f_2'.\mathsf{ff}.\mathbf{W}_1 = \begin{bmatrix} \mathbf{0} & \mathbf{0} \\ \mathbf{0} & f_2.\mathsf{ff}.\mathbf{W}_1 \end{bmatrix} \qquad f_2'.\mathsf{ff}.\mathbf{b}_1 = \begin{bmatrix} \mathbf{0} \\ f_2.\mathsf{ff}.\mathbf{b}_1 \end{bmatrix}$$

$$f_2'.\mathsf{ff}.\mathbf{W}_2 = \begin{bmatrix} \mathbf{0} & \mathbf{0} \\ \mathbf{0} & f_2.\mathsf{ff}.\mathbf{W}_2 \end{bmatrix} \qquad f_2'.\mathsf{ff}.\mathbf{b}_2 = \begin{bmatrix} \mathbf{0} \\ f_2.\mathsf{ff}.\mathbf{b}_2 \end{bmatrix}$$

Then, by Lemma 8.2, stack $f_1'$ on top of $f_2'$, or the other way around. (If we had multi-head attention, we could have combined $f_1$ and $f_2$ into a single layer.) $\qquad\square$

# 9 Putting It All Together: Example Constructions

In this section, we demonstrate how our tools can be applied to construct a transformer for certain tasks. In particular, we present the construction of induction heads, as well as a construction to recognize the Dyck language adapted from Bhattamishra et al. (2020); Yao et al. (2021); Yang et al. (2024). We note that the constructions presented here require multi-layer Transformers. Communication complexity-based arguments show that, for sequences of length $N$, any single-layer Transformer whose size is $o(N)$ or independent of $N$ cannot compute induction heads (Sanford et al., 2024b) or recognize Dyck languages (Bhattamishra et al., 2024).

## 9.1 Induction Heads

The term *induction head* was coined by Elhage et al. (2021) in order to describe a transformer which performs a basic pattern-recognition task: If the current symbol is $A$ and the previous occurrence of $A$ was followed by a $B$, then predict the next symbol as $B$. In other words, the induction head is formally specified as a function with the following type:

$$\mathsf{Induct} \colon \Sigma^+ \to \Sigma^+$$

There are a few variants of the induction head, each of which can be solved in a slightly different way.

### 9.1.1 Most-Recent Induction

In this case, the given sequence may contain multiple previous instances of the symbol to perform induction with. Here, we take the right-most instance for the induction. If none exists, we predict the current symbol. For example in the sequence, at the last position we will predict $C$, because the last symbol is $A$ and the symbol following the second-to last $A$ is $C$. The same applies at other positions.

$$\mathsf{Induct}_{\mathsf{rightmost}}(ACABDACDCA) = ACCBDBAADC$$

Then, the predicted symbol is $C$. This mechanism can be achieved using the following construction, using the input above as a running example. We will represent each symbol with its one-hot encoding $\mathbf{e}_\sigma$ for $\sigma \in \Sigma$.

- First, we use a one-hot embedding in order to encode the sequence

$$\mathsf{WE}(ACABDACDCA) = \mathbf{e}_A, \mathbf{e}_C, \mathbf{e}_A, \mathbf{e}_B, \mathbf{e}_D, \mathbf{e}_A, \mathbf{e}_C, \mathbf{e}_D, \mathbf{e}_C, \mathbf{e}_A$$

- For every symbol, use the routing construction from Lemma 8.1 and the pred construction from Section 5.4 to retrieve the embedding of the predecessor symbol (returning 0 when none exists).

$$\mathsf{Pred}(ACABDACDCA) = \mathbf{0}, \mathbf{e}_A, \mathbf{e}_C, \mathbf{e}_A, \mathbf{e}_B, \mathbf{e}_D, \mathbf{e}_A, \mathbf{e}_C, \mathbf{e}_D, \mathbf{e}_C$$

- If the current symbol is $A$, we can perform a similar construction as the one-hot encoding lookup from Section 5.3.1 in order to retrieve the right-most position whose predecessor is $A$. In this case, we use $\mathbf{q}_i = \mathsf{WE}(x_i)$, $\mathbf{k}_j = \mathsf{Pred}(x_j)$, and $\mathbf{v}_j = \mathsf{WE}(x_j)$. Attention will be maximized at all positions with the same symbol as $x_i$. Then, right-most unique hard attention (say, using Section 5.6) can be used to maximize attention on the rightmost of these positions, and $\mathbf{W}^{(V)}$ retrieves the symbol at that position.

$$\mathsf{Retrieve}(ACABDACDCA) = \mathbf{e}_A, \mathbf{e}_C, \mathbf{e}_C, \mathbf{e}_B, \mathbf{e}_D, \mathbf{e}_B, \mathbf{e}_A, \mathbf{e}_A, \mathbf{e}_D, \mathbf{e}_C$$

This construction also could have been achieved using soft-attention. Because the dot product of one-hot encoding vectors always results in 0 or 1, we have a gap of $\gamma = 1$ between the largest and second-largest attention scores. Thus, one could use a temperature scaling factor of $\tau = \frac{1}{\log 8N}$ as described in Section 5.7 to simulate average-hard attention using softmax attention, and then use a positional encoding including the term $\frac{-1}{j}$ to simulate right-most hard attention using average-hard attention as described in Section 5.6.

### 9.1.2 Most-Frequent Induction

In this case, the given sequence may contain multiple previous instances of the symbol to perform induction with. Here, we take the most frequently occurring induction as the ultimate induction. For example in the sequence, we perform induction at the last position with the symbol $A$, which occurs three times previously. Here, the predicted symbol is $C$, because $AC$ occurs twice while $AB$ only occurs once. When two symbols have the same probability, we can break ties arbitrarily since there are finitely many symbols (here, we can use the alphabetic ordering to break ties).

$$\mathsf{Induct}_{\mathsf{frequent}}(ACABDACDCA) = ACCBDBAAAC$$

This mechanism can be achieved using the following construction

- First, we use a one-hot embedding in order to encode the sequence

$$\mathsf{WE}(ACABDACDCA) = \mathbf{e}_A, \mathbf{e}_C, \mathbf{e}_A, \mathbf{e}_B, \mathbf{e}_D, \mathbf{e}_A, \mathbf{e}_C, \mathbf{e}_D, \mathbf{e}_C, \mathbf{e}_A$$

- For every symbol, use the routing construction from Lemma 8.1 and the pred construction from Section 5.4 to retrieve the embedding of the predecessor symbol (returning 0 when none exists). This allows us to encode the bigram $x_{i-1}x_i$ at each position.

$$\mathsf{Bigram}(ACABDACDCA) = \begin{bmatrix} \mathbf{e}_A \\ \mathbf{0} \end{bmatrix}, \begin{bmatrix} \mathbf{e}_C \\ \mathbf{e}_A \end{bmatrix}, \begin{bmatrix} \mathbf{e}_A \\ \mathbf{e}_C \end{bmatrix}, \begin{bmatrix} \mathbf{e}_B \\ \mathbf{e}_A \end{bmatrix}, \begin{bmatrix} \mathbf{e}_D \\ \mathbf{e}_B \end{bmatrix}, \begin{bmatrix} \mathbf{e}_A \\ \mathbf{e}_D \end{bmatrix}, \begin{bmatrix} \mathbf{e}_C \\ \mathbf{e}_A \end{bmatrix}, \begin{bmatrix} \mathbf{e}_D \\ \mathbf{e}_C \end{bmatrix}, \begin{bmatrix} \mathbf{e}_C \\ \mathbf{e}_D \end{bmatrix}, \begin{bmatrix} \mathbf{e}_A \\ \mathbf{e}_C \end{bmatrix}$$

- Using the uniform attention construction from Section 5.1.2 and an appropriately define value matrix to select the desired bigram embedding, we can count the occurrences of each bigram $\sigma_1\sigma_2$ up to each position.

$$\text{BigramCount}_{AC}(ACABDACDCA) = 0, 1, 1, 1, 1, 1, 2, 2, 2, 2$$

- Using a comparison as described in Section 4.7, the symbol that most frequently follows an $A$ can be detected, and used as the prediction. At position $i$, we enter a case for the symbol $w_i = \sigma$. In this case, we look at every count $\text{BigramCount}_{\sigma\sigma'}$, and check for each $\sigma'$ if $\bigwedge_{\sigma''\neq\sigma'}\text{BigramCount}_{\sigma\sigma'}(w)_i \geq \text{BigramCount}_{\sigma\sigma''}(w)_i$. We return the alphabetically first $\sigma'$ for which this is true.

$$\text{Retrieve}(ACABDACDCA) = \mathbf{e}_A, \mathbf{e}_C, \mathbf{e}_C, \mathbf{e}_B, \mathbf{e}_D, \mathbf{e}_B, \mathbf{e}_A, \mathbf{e}_A, \mathbf{e}_A, \mathbf{e}_C$$

## 9.2 The Dyck Languages

The Dyck languages are important because they exemplify *hierarchical structure*, a key feature of both natural and formal languages. Recognizing them requires tracking long-distance dependencies and nested relationships, making them an excellent case study for a model's capacity for fundamental computational primitives. In Section 9.2.1 we present a construction to recognize DYCK-1, the prototypical Dyck language. DYCK-1 consists of well-nested strings from the alphabet $\Sigma = \{(,)\}$ and is defined by two conditions:

- PROPERTY 1 (Prefix Condition): In every prefix of the string, the count of open brackets must be greater than or equal to the count of close brackets.

- PROPERTY 2 (Balance Condition): The total count of open and close brackets in the entire string must be equal.

For example, the string ()(()) is in DYCK-1, while ())(() is not. Formally, DYCK-1 is the language generated by the grammar:

$$S \to \varepsilon \mid (S)S$$

The decision problem is to determine if an input string $w \in \{(,)\}^*$ is a member of DYCK-1. We present a transformer construction to recognize DYCK-1 in Section 9.2.1. In sections Sections 9.2.2 and 9.2.3 we introduce *depth-bounded* variants of the Dyck language and present two additional transformer constructions.

### 9.2.1 A Transformer Construction for Dyck-1

We now present a recipe to construct a transformer that recognizes DYCK-1, based on work by Bhattamishra et al. (2020). This construction requires the following ingredients: average-hard attention, no layer normalization, and (non-strict) future masking. The construction presented requires two layers, where we assume the use of residual connections for both the self-attention and feedforward components. We note that in practice, transformers learn to simulate this algorithm when trained from data (Bhattamishra et al., 2020).

**Intuition** A simple algorithm for recognizing DYCK-1 provides intuition for the transformer construction: we make a single left-to-right pass and maintain a running count of the number of open minus closed parentheses. As illustrated in Fig. 2, this count must be non-negative at each position, and be exactly zero at the final position of the string. The main idea is to use future-masked uniform attention (i.e., future-masked average hard attention) to compute this value at each position.

**Overview** For an input string $w$ of length $n$ and a position $0 \leq i < n$, let $O_i$ denote the number of open parentheses ('(') appearing in the prefix $w_{:i}$. Similarly, let $C_i$ denote the number of close parentheses (')') appearing in $w_{:i}$. We use a two-layer transformer to implement recognition of DYCK-1: The first layer's attention head computes the running balance $B_i = O_i - C_i$ and its feed-forward network calculates a prefix error signal $E_i$; then, the second layer's attention head aggregates the $E_i$ to compute the total error $t_n$. Finally, we inspect the output at the last position (index $n$) to check whether the string belongs in DYCK-1.

$$O_4 = 3, C_4 = 1, O_4 \geq C_4 \quad \checkmark \qquad O_3 = 1, C_3 = 2, O_3 < C_3 \quad \textcolor{red}{\times} \qquad O_3 = 2, C_3 = 1, O_3 \geq C_3 \quad \checkmark$$

$$( \ ) \ ( \ ( \ ) \ ) \qquad\qquad ( \ ) \ ) \ ( \ ( \qquad\qquad ( \ ) \ ( \ ( \ ($$

$$O_6 = 3, C_6 = 3, O_6 = C_6 \quad \checkmark \qquad O_6 = 3, C_6 = 3, O_6 = C_6 \quad \checkmark \qquad O_6 = 4, C_6 = 2, O_6 \neq C_6 \quad \textcolor{red}{\times}$$

Figure 2: Prefix-sum checks for membership in the Dyck language. Each subpanel plots the running counts $O_k$ (opens) and $C_k$ (closes) at each position $k$ of a candidate string of length $N = 6$. (Left) Valid Dyck-1 string: $O_k \geq C_k$ for all $k < N$ and $O_N = C_N$, satisfying non-negativity and balanced counts, respectively. (Center) Prefix violation: Although $O_N = C_N$, at position $k = 3$, we have $O_3 < C_3$, violating non-negativity. (Right) Here $O_k \geq C_k$ for all $k$, but $O_N \neq C_N$, so the total counts are unbalanced.

**Step 1: Running balance computation**  We first convert the symbolic inputs into a numerical format by embedding the '(' and ')' as follows:

$$x_i = \begin{bmatrix} o_i \\ 0 \\ 0 \\ 0 \end{bmatrix}, \quad o_i = \begin{cases} +1 & \text{if position } i \text{ is } (, \\ -1 & \text{if position } i \text{ is } ), \end{cases} \tag{41}$$

yielding a sequence of embedding vectors $\mathbf{x} = (x_1, \ldots, x_n)$. The core of the algorithm is to compute the running balance of parentheses at every position, which we do with future-masked uniform attention. In particular, we use the $(\mathsf{Avg}_\leftarrow)$ construction from Section 5.1.2 with a first-coordinate projection on $\mathbf{x}$ to compute the cumulative average:

$$(\mathsf{Avg}_\leftarrow(\mathbf{o}))_i = \frac{1}{i} \sum_{j=1}^{i} o_j = \frac{O_i - C_i}{i} = \frac{B_i}{i}, \quad \text{where } \mathbf{o} = (o_1, o_2, \ldots, o_n). \tag{42}$$

The running balance $B_i/i$ is then written into a workspace dimension (see Eq. (44)). Note that scaling the balance $B_i$ by a positive scalar $1/i$ does not change the sign of the running balance, so the necessary information to check the Dyck properties is preserved.

**Step 2: Computing balance violations**  Having used the first self-attention layer to compute a running balance at each position, we next aggregate an error signal into the final position, such that the final DYCK-1-acceptance check needs to only inspect the $n$-th position output. To do this, we use an FFN with weights set so as to compute $E_i = \mathrm{ReLU}(-B_i/i)$, noting that $E_i > 0$ iff the non-negativity condition of the running balance (PROPERTY 1) is violated. This can be achieved using an FFN with $\mathbf{W}_1 = \mathbf{W}_2 = \mathbf{I}$ and left and right matrices as in Section 8.2 used to negate the input and zero-out all output dimensions besides the third row.

**Step 3: Aggregating to the last position**  Finally, we use the second layer's self-attention to aggregate the error signal as follows:

$$t_i = (\mathsf{Avg}_\leftarrow(\mathbf{E}))_i = \frac{1}{i} \sum_{j=1}^{i} E_j, \quad \text{where } \mathbf{E} = (E_1, E_2, \ldots, E_n), \tag{43}$$

where note that at the final position, $t_n > 0$ iff the running balance (PROPERTY 1) has been violated at any position, and that $t_n = 0$ otherwise. Altogether, this construction performs the following operations at each position $i$

$$\begin{bmatrix} o_i \\ 0 \\ 0 \\ 0 \end{bmatrix} \xrightarrow{\text{Compute Balance}} \begin{bmatrix} o_i \\ B_i/i \\ 0 \\ 0 \end{bmatrix} \xrightarrow{\text{Negative } B_i \text{ Error?}} \begin{bmatrix} o_i \\ B_i/i \\ E_i \\ 0 \end{bmatrix} \xrightarrow{\text{Sum of Errors}} \begin{bmatrix} o_i \\ B_i/i \\ E_i \\ t_i \end{bmatrix} \tag{44}$$

The second layer's accumulation reduces our work to checking the output vector at the final position $n$. In particular, to check PROPERTY 1, it suffices to check that $t_n = 0$, since $t_n$ aggregates the prefix violations for all prefixes of the string. To check PROPERTY 2, it suffices to check that $B_n/n = 0$ (i.e., the final balance is zero). To implement these checks, however, we must first decide whether our construction's parameters should depend on the input length $n$ (see the discussion of *uniformity*, Section 2.3). In our setting, a *nonuniform* construction can output a $0/1$ binary value to indicate acceptance, while a *uniform* construction requires the final acceptance check to be performed externally.

**Uniform implementation** In a uniform implementation, the transformer's parameters are fixed and do not depend on the input length $n$. This approach is often preferred in theoretical analyses as it reflects a single model that can handle inputs of any length. Since a uniform model cannot use length-dependent comparison functions from Section 4.7, it cannot produce a single $0/1$ output. Instead, the acceptance condition is defined by an external check on the final output vector, in the manner described above.

**Nonuniform implementation** However, if one wishes to coerce the acceptance decision into a single $0/1$ value (at a particular position in the transformer's output) one must perform a nonuniform construction. To see this, consider the family of FFNs $\mathsf{EqZero}_\epsilon$ (which checks equality with zero) described in Section 4.7. A $\mathsf{EqZero}_\epsilon$ FFN is parameterized by $\epsilon$, the magnitude needed for a value to be considered different from zero. In our setting, using $\mathsf{EqZero}_{1/n^2}$ where $n$ is the length of the input sequence would suffice. This is because the smallest nonzero magnitude of the (scaled) running balance $B_i$ is $1/i$, and $i$ is upper-bounded by the input sequence length $n$. Since $t_n$ is computed by using $\mathsf{Avg}_\leftarrow$ to average the values of $B_i$, the smallest nonzero magnitude of $t_n$ is $1/n^2$. Similar reasoning shows that the smallest nonzero magnitude of $B_n/n$ is $1/n$.

One could add an additional transformer layer that uses an FFN (built from the components $\mathsf{EqZero}_\epsilon$ and logical AND from Section 4.8) to render a $0/1$ decision, however this would require fixing an upper bound on the maximum length of the input sequence that the transformer can process.

### 9.2.2 A Transformer Construction for Dyck-1-2

We will now present a construction that recognizes a *depth-bounded* Dyck language. In particular, we will focus on DYCK-1-2, which is the Dyck language formed with one kind of bracket in which the maximum nesting depth is 2. The maximum nesting depth of a string being $D$ means that the total number of unclosed brackets in any prefix is at most $D$. This construction for DYCK-1-2 appears in Yang et al. (2024) and is based on Yao et al. (2021). This construction requires the following ingredients: (unique or averaging) hard attention, no layer normalization, strict future masking, strict past masking, and $i/n$ positional encodings.

**Intuition** Consider the input string ((())(()()), which is a member of DYCK-1-2 (and thus should be accepted). We will check membership in two steps. First, we will look at pairs of adjacent brackets and mark all the matching pairs. In our example string, we would mark "(())(()())". This suffices to check for depth-one matches.

In the second step, we filter out all the brackets we marked in the first step and then mark all the matching adjacent pairs in this filtered string. This checks for depth-two matches. To continue our example, filtering out all the brackets we marked in step 1 leaves us with the string "()()." In this case, each bracket matches an adjacent bracket in the filtered string, so we would mark all the remaining brackets.

In the final step, we accept the string iff all brackets have been marked as matched after the second step.

**Overview** Our description of this construction will pull together many ingredients from earlier sections of the cookbook. The main idea of this construction is to use multiple transformer layers to perform steps 1 and 2 in sequence. Importantly, we need to keep track of which brackets have been marked as "matched"

during steps 1 and 2. We do this by maintaining a "active bit" within the intermediate representation at each position. Concretely, we initialize our embedding vector $x_i$ at each position $i$ as follows:

$$x_i = \begin{bmatrix} o_i \\ 1 \\ 0 \\ 0 \\ i/n \end{bmatrix}, \quad o_i = \begin{cases} +1 & \text{if position } i \text{ is (,} \\ -1 & \text{if position } i \text{ is ).} \end{cases} \tag{45}$$

As before, let $o_i$ denote whether position $i$ is open $(+1)$ or closed $(-1)$. The second entry is the "active bit", where 1 denotes that the current position is not yet matched, and which we later set to 0 to denote a successful match. The third and fourth entries are initialized to zero and serve as scratch space on which to store intermediate computations. Finally, we use a positional encoding containing $i/n$.

**Step 1: Identifying depth-1 matches**  We first use the predecessor construction in Section 5.4 that uses strict future masking, as well as the successor construction (a slight modification of predecessor), to fetch the bracket types immediately to the left and right of the current position. Based on this, we then use an FFN to check for depth-1 matches and update the "active-bit" as follows:

$$\begin{bmatrix} o_i \\ 1 \\ 0 \\ 0 \\ i/n \end{bmatrix} \xrightarrow{A_{\text{left}}, A_{\text{right}}} \begin{bmatrix} o_i \\ 1 \\ o_{i-1} \\ o_{i+1} \\ i/n \end{bmatrix} \xrightarrow{\text{FFN Mark Matched}} \begin{bmatrix} o_i \\ a_i \\ 0 \\ 0 \\ i/n \end{bmatrix} \tag{46}$$

A value of $a_i = 1$ indicates that no depth-1 match has occurred, such as when $o_{i-1}o_io_{i+1}$ corresponds to the bracket sequence )((. In particular, let:

$$a_i = \begin{cases} 0 & \text{if } o_i = +1 \text{ and } o_{i+1} = -1, \\ 0 & \text{if } o_i = -1 \text{ and } o_{i-1} = +1, \\ 1 & \text{otherwise,} \end{cases} \tag{47}$$

which may be implemented using techniques from Section 4. As a corner case, let $o_{-1} = o_{n+1} = 0$, or some other distinguished value, which the FFN's conditional implementation can handle.

**Step 2: Identifying depth-2 matches**  The depth-2 match is similar to above, except at each position we search for the nearest left and right components that are not yet matched (i.e., whose active bit is still 1). Finding the nearest not-yet-matched position to the left can be done similarly to the predecessor construction employed above, and may be done by by running strict future-masked UHAT on the following scores:

$$\text{UHAT}\left(\frac{1}{n} + a_1, \frac{2}{n} + a_2, \ldots, \frac{n}{n} + a_n\right)_i \quad \text{for } i = 1, \ldots, n, \tag{48}$$

We add the active bit $a_j$ to the positional encoding $\frac{j}{n}$ at position $j$ to ensure that the maximal attention corresponds to an unmatched position. These scored may be achieved by applying a linear transform to the embedding state of Eq. (46), such as via the self-attention's $W^{(Q)}$ and $W^{(K)}$ matrices. A similar approach (using strict past masking) allows us to concentrate attention on the nearest unmatched position to the right. We then proceed analogously to step 1, where let:

$$\begin{bmatrix} o_i \\ a_i \\ 0 \\ 0 \\ i/n \end{bmatrix} \xrightarrow{\text{Select Nearest Active}} \begin{bmatrix} o_i \\ a_i \\ l_i \\ r_i \\ i/n \end{bmatrix} \xrightarrow{\text{Check Match}} \begin{bmatrix} o_i \\ a_i' \\ 0 \\ 0 \\ i/n \end{bmatrix} \tag{49}$$

where $l_i, r_i \in \{\pm 1\}$ correspond to the left and right tokens not yet matched. Likewise, let $a'_i = 1$ if position $i$ is not part of any depth-1 or depth-2 matches, and let $a'_i = 0$ otherwise if it is matched. As above, these conditional checks and updates can be implemented via an FNN.

**Step 3: Check for unmatched brackets** Finally, it remains to check that no positions remain unmatched: we accept the string iff at all positions we have $a'_1 = \cdots = a'_n = 0$. This is similar to the zero-check in our earlier DYCK-1 construction, and similar discussions on uniformity vs. non-uniformity apply.

### 9.2.3 Generalization to Dyck-k-D

The DYCK-$k$ language consists of well-balanced strings over an alphabet of $k$ parenthesis pairs $(_i, )_i$, for $i \in [k]$, generated by the context-free grammar:

$$S := \varepsilon \mid (_i S)_i S, \quad \text{for } i \in [k].$$

The DYCK-$k$-$D$ language is the subset of DYCK-$k$ where the maximum nesting depth (the number of unclosed open parentheses at any point in the string) does not exceed $D$.

We now generalize the construction of Section 9.2.2 to DYCK-$k$-$D$, where the algorithmic idea is to find and cancel matching parentheses iteratively for depths $1, \ldots, D$. To do this, we first expand the parentheses encoding from Eq. (45) as:

$$o_i \in \{-k, \ldots, -1, +1, \ldots, +k\},$$

allowing us to encode both the openness $(\pm)$ and type $(1, \ldots, k)$ of the parenthesis at position $i$. Then, we repeat the select-nearest-active and check-match of Eq. (49) $D$ times, where we augment the feedforward component of Eq. (46) to handle matching for $k$ different parentheses pairs. Finally, the acceptance criterion is the same as Step 3 above, which involves checking whether any positions remain active (unmatched). As a final remark, we note that generalizations to DYCK-$k$ of unbounded depth were discussed by Yao et al. (2021, Sec B.4) and Yang et al. (2026, Sec D).

## 10 Discussion

This cookbook presents an overview of what transformers can compute and how they may compute. The recipes presented herein give a *mise en place* of ingredients sampled from the literature. Theoretically, they are a useful reference for investigations in transformer expressivity and learnability. Practically, they are a starting point for experiment design in areas such as mechanistic interpretability and architecture design. We have provided a code repository containing implementations of each construction to support future work. However, our curation is by no means exhaustive: new ideas and techniques are under active development. Indeed, we eagerly await what new flavors the reader will discover!

## 11 Acknowledgments

We would like to thank the anonymous reviewers for their helpful comments. We would like to thank the following chefs for bringing new flavors to our kitchen: Lena Strobl, Yuval Pinter, Dana Angluin, Jonathan Rawski, Gail Weiss, Brian DuSell, Marco Cognetta, Gabriel Faria, Travis Bartley, Nithila Sivapunniyam, Yingshan Chang, Akos Kadar, Shahaf Bassan, Yash Sarrof, Frederick Morlock, and other members of the FLaNN Discord Server.

This material is based upon work supported by the National Science Foundation under Grant Nos. DMS-2502292, CCF-2505865, CCF-2313010, and IIS-2331783, the ARPA-H program on Safe and Explainable AI under the grant D24AC00253-00, and a gift from AWS AI to the ASSET Center at Penn. Merrill is supported a Two Sigma PhD fellowship, an NSF Graduate Research Fellowship, and the Allen Institute for Artifical Intelligence, Svete is supported by the ETH Zürich AI Center doctoral fellowship, and Yang is supported by an NSF Graduate Research Fellowship under Grant No. 2236418.

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
