# OpenReview forum: "The Transformer Cookbook"
_TMLR — Accepted by TMLR_

### Review · Reviewer_QZtd · 2025-10-21

**Summary Of Contributions:**

In this paper, authors Unifies scattered “hand-crafted transformer” results into a single, accessible cookbook that directly encodes algorithms in transformer parameters.

**Audience:**

Yes

**Audience Explanation:**

Given TMLR’s audience of transformer researchers, this submission would be a valuable reference for the community.

**Broader Impact Concerns:**

Na.

**Claims And Evidence:**

Yes

**Claims Explanation:**

Although no new theory or results are presented, the paper’s well-organized consolidation of existing methods serves as a useful reference for researchers.

**Requested Changes:**

Overall, I have no major concerns about the submission. In my view, it is acceptable as is and would serve as a useful reference for researchers in the transformer community.

That said, the paper would benefit from an explicit, self-contained summary of the model architecture: ideally a short “Model at a Glance” subsection placed before each technical section. For example, you model piecewise-linear behavior via ReLU, while multiplicative interactions are approximated using a Taylor expansion of the GELU activation. Please clarify whether these two mechanisms are intended to be complementary or if they introduce overlapping, or potentially conflicting effects.

Finally, there are a few typos. Please proofread the submission carefully and revise. For instance, on page 31, “Thr prototypical Dyck language” should read “The prototypical Dyck language.”

---

> ### Author Response · Authors · 2025-11-21
> **Author Response**
>
> # Response to Reviewer QZtd
>
> ## Requested Changes
>
> We thank the reviewer for their positive reception and feedback. We address the main points below. Additionally, we refer the reviewer to our revised manuscript, where the major changes are highlighted.
>
>
> >That said, the paper would benefit from an explicit, self-contained summary of the model architecture: ideally a short “Model at a Glance” subsection placed before each technical section. For example, you model piecewise-linear behavior via ReLU, while multiplicative interactions are approximated using a Taylor expansion of the GELU activation. Please clarify whether these two mechanisms are intended to be complementary or if they introduce overlapping, or potentially conflicting effects.
>
>
> Thank you for the suggestions on how to improve our exposition. We are implementing your ideas as follows. First, we have expanded the exposition in Section 2.2, which overviews the transformer architecture, to provide a better roadmap of the remaining chapters. We also added an additional column to Table 2, showing assumptions on positional encoding for our self-attention constructions. Finally, we have clarified several architectural assumptions of specific components in later chapters where appropriate (see: new discussions on GELU).
>
>
> Our rationale is this: because transformer components are highly customizable, we felt it appropriate to present a more high-level architectural view at the beginning of the manuscript, while deferring highly specific dependency details to later. We acknowledge that a problematic scenario occurs if one wishes to use two constructions whose assumptions are incompatible. We hope that our cookbook, and the references therein, will provide a jumping-off-point for researchers to adjust details of existing "recipes" to create new constructions.
>
>
>
> >Finally, there are a few typos. Please proofread the submission carefully and revise. For instance, on page 31, “Thr prototypical Dyck language” should read “The prototypical Dyck language.”
>
>
> Thank you for catching these. We have fixed them, as well as a few other typos.

---

### Review · Reviewer_ybd8 · 2025-11-06

**Summary Of Contributions:**

The content of the paper is exactly what the abstract promises.
An overview of functions which can be either exactly implemented or very well approximated (square function) by a transformer.

**Audience:**

Yes

**Audience Explanation:**

The intention of the paper is to ease an entry for people interested in which function transformers can implemented exactly. For those people, this is very good. I also believe this to be quite good for researchers interested in extracting functions (computational graphs) from transformers. The last type of audience I can imagine are those, who are interested in learning algorithms and functions. The paper can tell them if the algorithm / function to be learned can be represented by transformer exactly. Personally, I have found this type of thinking frequently important for choosing the right architecture for my problem.

**Broader Impact Concerns:**

I do not see any ethical concern. I think that broader impact would be limited, mainly because the paper summarizes one aspect of understanding transformers, which few people are interested (this is not a criticism, this is statement of the fact that most people are interested in great applications).

**Claims And Evidence:**

Yes

**Claims Explanation:**

The paper is mostly about synthesis of the literature and as such, the claims are convincing. I admit that I have not checked the references, but I was looking at the arguments of how functions are implemented and they seems to be fine.

**Requested Changes:**

I am not an expert on this topic, but I have found weird that RASP language [1] is not mentioned. For me, this was one of the first papers trying to formalize, which operations transformers can do. There is a companion paper TRACR [2], which might be useful as well.

Although the paper is very long, I would like to also see examples of use. Why knowing which types of function transformers implements is important and what can I do with it? I have hinted something above, but I would expect to learn, where and how this is used. Does it have any practical impact, or it is mostly just a theoretical work.


[1] Weiss, G., Goldberg, Y., & Yahav, E. (2021, July). Thinking like transformers. In International Conference on Machine Learning (pp. 11080-11090). PMLR.

[2] Friedman, Dan, Alexander Wettig, and Danqi Chen. "Learning transformer programs." Advances in Neural Information Processing Systems 36 (2023): 49044-49067.

[3]  Lindner, D., Kramár, J., Farquhar, S., Rahtz, M., McGrath, T., & Mikulik, V. (2023). Tracr: Compiled transformers as a laboratory for interpretability. Advances in Neural Information Processing Systems, 36, 37876-37899.

---

> ### Author Response · Authors · 2025-11-21
> **Author Response**
>
> # Response to Reviewer ybd8
>
> ## Requested Changes
>
>
> We thank the reviewer for their positive reception and feedback. We address the main points below. Additionally, we refer the reviewer to our revised manuscript, where the major changes are highlighted.We thank the reviewer for their positive assessment. We address the main points below.
>
> > I am not an expert on this topic, but I have found weird that RASP language [1] is not mentioned. For me, this was one of the first papers trying to formalize, which operations transformers can do. There is a companion paper TRACR [2], which might be useful as well.
>
>
> Thank you for catching this, these are indeed important references! We have accordingly added them with relevant discussion in the introduction.
>
>
> > Although the paper is very long, I would like to also see examples of use. Why knowing which types of function transformers implements is important and what can I do with it? I have hinted something above, but I would expect to learn, where and how this is used. Does it have any practical impact, or it is mostly just a theoretical work.
>
>
> Knowing which functions a transformer can implement helps us understand its fundamental strengths and limitations [1]. A major application is in the derivation of theoretical insights that can explain empirical observations and guide future development. For instance, [2] shows that chain-of-thought reasoning provably improves the expressive power of transformers, thereby providing theoretical foundations to a long-observed empirical phenomena. Another notable work is [3], which shows that width-scaling in transformers alone is insufficient to solve long sequential tasks (e.g., code execution/comprehension), and that adding depth or other mechanisms like chain-of-thought are also needed for performance. Additionally, [4] argues that transformers exploit parallel computation to solve tasks that earlier architectures cannot, which has implications towards what tasks transformers can be expected to perform well on. Theoretical results like the above rely on the constructions herein, and serve as important heuristics in high-cost engineering endeavors like architecture design, data selection, and training runs.
>
> With the final draft we are providing a suite of generative unit tests for each construction (using numpy, scipy and hypothesis + pytest), with cross-references to the appropriate sections. The default number of examples is 100 per construction but can be changed to a higher number for more robustness.
>
> [1] Strobl et al. What formal languages can transformers express? a survey. TACL 2024.
>
> [2] Merrill et al. The Expressive Power of Transformers with Chain of Thought. ICLR 2024.
>
> [3] Liu et al. Transformers Learn Shortcuts to Automata. ICLR 2023.
>
> [4] Sanford et al. Transformers, parallel computation, and logarithmic depth. ICML 2024.

---

> > ### Comment · Reviewer_ybd8 · 2025-11-26
> > **I recommend the paper for acceptance**
> >
> > I am satisfied with answers. I think that the paper will be interesting for some researchers.

---

### Review · Reviewer_rSer · 2025-11-13

**Summary Of Contributions:**

The paper's primary contribution is to serve as a unified cookbook that synthesizes the fragmented literature on programming transformers. It collects and systematizes techniques for building specific, pre-defined algorithms directly into a transformer's parameters. It provides a library of concrete recipes including explicit weight matrices and bias vectors for implementing components like arithmetic, conditional logic, and various forms of attention-based table lookup. The paper concludes by showing how to assemble these components to solve more complex tasks, using induction heads and Dyck-language recognition as examples.

**Key Strengths**:

- Synthesis and Clarity: The paper's main strength is its value as a single, comprehensive reference. It pulls together dozens of techniques from disparate papers into one consistent notational framework, which is a significant service to the community.

- Concrete and Actionable: The "recipe" format is excellent for teaching and reproducibility. Providing explicit parameter matrices makes the constructions tangible and removes ambiguity.

- Breadth: The cookbook is thorough, covering a wide range of useful primitives from FFN-based computation and techniques using Layer Normalization that are useful for practitioners and researchers.

**Key Weaknesses**:

- Idealized Assumptions: Many core recipes rely on non-standard or impractical assumptions, such as "average-hard attention" (e.g., the Dyck-1 construction in the Examples section), setting LayerNorm epsilon to zero (LN.ϵ=0) (see the Layer Normalization section), or requiring architectural changes like "selective" LayerNorm. These assumptions are interesting but depart from standard transformer architectures and deserve clearer discussion of practicality.

- Missing Practical Validation: The paper claims to be a practical cookbook but provides no empirical validation or code. It's unclear if these recipes, particularly their soft-attention approximations, work as described under real-world constraints. The paper is a collection of blueprints that often point to prior work for details, but the authors do not provide implementations or experiments that demonstrate the constructions are stable in practice, which one would expect from a cookbook.

**Additional Comments:**

I want to reiterate that this is a highly useful and valuable piece of work. The "cookbook" metaphor is apt, and the authors have done a great service by synthesizing scattered results. The paper is well-organized and the concrete recipes are a strong contribution.

My request for some revisions is not intended to diminish the work. Rather, it is intended to push the paper to fulfill its ambitious goal of being a practical cookbook. As presented, it is closer to an idealized theoretical cookbook. By providing code, a table of assumptions, and a practical analysis of its soft-attention approximations, the paper can be elevated to a very strong submission. I look forward to the revised version. The paper is very long and I have tried my best to follow it with my understanding; I hope these comments help the authors improve clarity, reproducibility, and practical applicability.

**Audience:**

Yes

**Audience Explanation:**

Yes, the TMLR audience, which sits at the intersection of theory and practice, would be extremely interested in this work.

- Interpretability Researchers: This community will find this paper invaluable. It provides a "library of idealized circuits" that can be used as formal hypotheses when trying to understand the mechanisms learned by real models (e.g., "Is this model performing table lookup using a mechanism similar to the 'almost-orthogonal' recipe?").

- Theorists: Researchers working on the formal expressive power of transformers will appreciate this paper as a clean, unified, and well-organized synthesis of dozens of scattered results. The consistent notation (Section 2) alone is a valuable contribution.

- Architecture Designers: The paper provides a clear functional analysis of how components like LayerNorm and positional encodings contribute to or limit a transformer's algorithmic capabilities, which could inform future architecture design.

The central problem of "what can transformers compute and how" is one of the most important in the field, and this paper provides a uniquely accessible and comprehensive entry point.

**Broader Impact Concerns:**

No. This is a foundational, theoretical, and mechanistic paper. It does not introduce a new model, dataset, or application with direct, foreseeable societal impacts.

**Claims And Evidence:**

Yes

**Claims Explanation:**

The paper's theoretical claims (i.e., "if you had an idealized machine with hard attention, these weights would produce this result") are well-supported by the mathematical constructions. However, the paper's implicit practical claim that this is a useful "cookbook" for understanding and building algorithms in real transformers is not supported throughly by convincing evidence. The evidence is missing in three key areas:

- Reliance on Impractical Primitives: Many constructions rely on components not present in standard transformers. For example, the Dyck-1 recipe explicitly requires average-hard attention. The layernorm-hash construction relies on LN.ϵ=0 and the ability to selectively apply LN, which the paper acknowledges would require architectural modifications. The amplification trick similarly assumes LN.ϵ=0. These constructions are theoretically interesting but do not, by themselves, establish practicality for standard models.

- Unquantified Approximations: The paper's primary method for bridging hard attention to soft attention is to scale queries (see the attention approximation discussion). It proposes a scaling factor τ = γ / ln(8N) and claims this can make approximation error small enough to round. This argument is presented qualitatively but lacks a worked numerical example or practical analysis of typical parameter regimes; the authors should spell out the numeric implications and provide explicit error bounds for representative N and γ values.

- The paper positions itself as a "cookbook," which implies practical applicability. However, as presented it is largely theoretical and lacks empirical validation, such as anonymized code artifacts or experiments, to demonstrate how these constructions perform in practice. It is unclear whether the recipes, particularly soft-attention approximations or those relying on idealized components, function as intended when implemented in common deep-learning libraries.

While the paper provides an excellent theoretical synthesis, the absence of basic implementations or verification makes it difficult to assess practical robustness. This gap between theory and practice undermines the paper's stated goal of providing a "foundation for... empirical investigations", since the first empirical verification step is missing.

**Requested Changes:**

- Add a Summary Table of Assumptions. This is the most critical change for clarity. The authors must add a table that lists every major recipe (e.g., "Index-Lookup: One-Hot," "Index-Lookup: LN-Hash," "Dyck-1") and, for each, lists its key properties:

        (i) Width / Param cost (e.g., O(N), O(logN), O(1))

        (ii) Dependence on N (uniform/non-uniform)

        (iii) Required Primitives (e.g., "Softmax," "Hard Attention," "Selective LN," "LN.ϵ=0")

        (iv) Error bound or "Practicality" (e.g., "Exact," "Approx. (see Sec 5.7)," "Theoretical-only")

- Provide a Reproducibility Artifact (Code). The authors should provide a small, runnable code release (e.g., a Python script with numpy/PyTorch) that implements a subset of the most important recipes. This will serve as the "convincing evidence" that is currently missing. A good test suite would be:
    (i) The "almost-orthogonal" index lookup
    (ii) The "quadratic maximization" index lookup
    (iii) The soft-attention approximation demonstration, showing empirical error after applying the proposed τ scaling
    (iv) The Dyck-1 construction using the soft-attention approximation, to measure accuracy on strings of increasing length

- Add Practical Numerical Analysis. For the soft-attention approximation, the authors should add a paragraph or appendix analyzing its practicality. Using their formula τ = γ / ln(8N), they should present worked numeric examples for representative N and γ values and discuss the resulting logits and whether the approximation gives sufficiently small error for rounding in those regimes.

- Formalize Key Claims: For the most important constructions (like the index-lookup methods and the soft-attention approximation), the claims should be stated as formal Propositions or Lemmas with all preconditions, assumptions, and error bounds made explicit. This is more rigorous than the current "recipe" text.

- Clarify Architectural Assumptions: The paper must be more explicit in the main text when a "recipe" departs from a standard, off-the-shelf transformer. The suggestion of adding a linear projection before LayerNorm is an architectural modification and must be clearly marked as such.

---

> ### Author Response · Authors · 2025-11-21
> **Author Response Part 1**
>
> # Response to Reviewer rSer
>
> ## Weaknesses
>
>
> > Idealized Assumptions: Many core recipes rely on non-standard or impractical assumptions, such as "average-hard attention" (e.g., the Dyck-1 construction in the Examples section), setting LayerNorm epsilon to zero (LN.ϵ=0) (see the Layer Normalization section), or requiring architectural changes like "selective" LayerNorm. These assumptions are interesting but depart from standard transformer architectures and deserve clearer discussion of practicality.
>
> > Reliance on Impractical Primitives: Many constructions rely on components not present in standard transformers ...
>
>
> Thank you for touching on this important point. We agree a more thorough discussion of these assumptions would make for a better writeup. First, we would like to note that all assumptions we use in the cookbook are standard assumptions used throughout the literature of transformer theory in the past years and are motivated by practice [6]:
>
> - (Average-Hard Attention). This assumption has a strong precedent in the literature, appearing in early seminal papers like [5] and more recent influential works like [4] and [1]. Average-hard attention is motivated by the observed behavior of transformers in practice, which often focus attention on a small subset of positions [3].
> - (LayerNorm). The assumptions we cite have precedent in the literature. For instance, [4] introduces the selective layernorm, and [7] use LN.ϵ=0 in their constructions. Furthermore, these theoretical results still make for good predictors of model behavior in practice; despite the assumptions, these results correctly predict that more CoT steps increase model capabilities [4] and that are able to predict length generalization in transformers [2].
>
> Nevertheless, it is still the case that we gain and lose something with each assumption. However, by collecting all these different constructions and their assumptions in one place, our writeup enables readers to see the entire landscape of assumptions and see the architectural implications of each one. For example, in Section 5.3, we put together four independently developed discovered constructions for performing a lookup operation, allowing readers to compare and contrast the various assumptions made in each one. We agree with you that we need to discuss this more, and we've expanded our manuscript, particularly in the introduction.
>
> [1] Barceló et al. Logical Languages Accepted by
> Transformer Encoders with Hard Attention. ICLR 2024.
>
> [2] Huang et al. A Formal Framework for Understanding Length Generalization in Transformers. ICLR 2025.
>
> [3] Merrill et al. Effects of parameter norm growth during transformer training: Inductive bias from gradient descent. EMNLP 2021.
>
> [4] Merrill & Sabharwal. The Expressive Power of Transformers with Chain of Thought. ICLR 2024.
>
> [5] Pérez et al. Attention is Turing-Complete. JMLR 2021.
>
> [6] Strobl et al. What formal languages can transformers express? a survey. TACL 2024.
>
> [7] Yang & Chiang. Counting Like Transformers: Compiling Temporal Counting Logic Into Softmax Transformers. COLM 2024.
>
> > Missing Practical Validation: The paper claims to be a practical cookbook but provides no empirical validation or code. It's unclear if these recipes, particularly their soft-attention approximations, work as described under real-world constraints. The paper is a collection of blueprints that often point to prior work for details, but the authors do not provide implementations or experiments that demonstrate the constructions are stable in practice, which one would expect from a cookbook.
>
> > The paper positions itself as a "cookbook," which implies practical applicability ...
>
> With the final draft we are providing a suite of generative unit tests for each construction (using numpy, scipy and hypothesis + pytest), with cross-references to the appropriate sections. The default number of examples is 100 per construction but can be changed to a higher number for more robustness.

---

> > ### Author Response · Authors · 2025-11-21
> > **Author Response Part 2**
> >
> > ## Missing Evidence
> >
> > > Unquantified Approximations: The paper's primary method for bridging hard attention to soft attention is to scale queries (see the attention approximation discussion). It proposes a scaling factor τ = γ / ln(8N) and claims this can make approximation error small enough to round. This argument is presented qualitatively but lacks a worked numerical example or practical analysis of typical parameter regimes; the authors should spell out the numeric implications and provide explicit error bounds for representative N and γ values.
> >
> > Thank you for the suggestion, and we agree this would be helpful to see. Because this construction is used to approximate specific hard-attention constructions using soft-attention, we decided to add a brief discussion of this into the Induction Heads example 9.1.1.
> >
> > We wrote a generative test for index-lookup (like for the rest of the contructions), that shows that the approximation works for a number different embedding dimensions and lengths within a 1e-6 tolerance. This will be included in the code artifact with the final draft.
> >
> >
> > ## Requested Changes
> >
> >
> > >Add a Summary Table of Assumptions. This is the most critical change for clarity. The authors must add a table that lists every major recipe (e.g., "Index-Lookup: One-Hot," "Index-Lookup: LN-Hash," "Dyck-1") and, for each, lists its key properties:
> >
> > We agree that making the assumptions clear is important for clarity. We found making one table with all assumptions would require too many columns to fit neatly on one page, so we elect to present assumptions as follows:
> > 1. The FFN constructions (table 1) all assume ReLU activations, with the exception of multiplication. We note this one exception in text and leave others unmarked.
> > 2. For self-attention constructions (table 2), we list the assumed kind of attention (SMAT, UHAT, and/or AHAT) as well as any positional encodings used. Since we present multiple index-lookup constructions, we detail their assumptions in table 5.
> > 3. For the higher-level constructions (e.g. "Dyck-1") there are many assumptions, and we have presented these in a new table 3.
> >
> >
> > > Provide a Reproducibility Artifact (Code). The authors should provide a small, runnable code release (e.g., a Python script with numpy/PyTorch) that implements a subset of the most important recipes. This will serve as the "convincing evidence" that is currently missing. A good test suite would be: (i) The "almost-orthogonal" index lookup (ii) The "quadratic maximization" index lookup (iii) The soft-attention approximation demonstration, showing empirical error after applying the proposed τ scaling (iv) The Dyck-1 construction using the soft-attention approximation, to measure accuracy on strings of increasing length
> >
> > With the final draft we are providing a suite of generative unit tests for each construction (using numpy, scipy and hypothesis + pytest), with cross-references to the appropriate sections. The default number of examples is 100 per construction but can be changed to a higher number for more robustness.
> >
> > > Add Practical Numerical Analysis. For the soft-attention approximation, the authors should add a paragraph or appendix analyzing its practicality. Using their formula τ = γ / ln(8N), they should present worked numeric examples for representative N and γ values and discuss the resulting logits and whether the approximation gives sufficiently small error for rounding in those regimes.
> >
> > Thank you for the suggestion, we agree this would be helpful to see. Because this construction is used to approximate specific hard-attention constructions using soft-attention, we decided to add a brief discussion of this into the Induction Heads example 9.1.1.
> >
> > > Formalize Key Claims: For the most important constructions (like the index-lookup methods and the soft-attention approximation), the claims should be stated as formal Propositions or Lemmas with all preconditions, assumptions, and error bounds made explicit. This is more rigorous than the current "recipe" text.
> >
> > Thank you for these suggestions. We will make sure to emphasize the preconditions and assumptions of the major constructions, such as the index-lookup methods and worked examples. That being said, we believe a more casual prose will make the recipes more accessible and applicable in different situations, and will opt to keep this out of most constructions for sake of readability.
> >
> > > Clarify Architectural Assumptions: The paper must be more explicit in the main text when a "recipe" departs from a standard, off-the-shelf transformer. The suggestion of adding a linear projection before LayerNorm is an architectural modification and must be clearly marked as such.
> >
> > Thank you for this suggestion, we have made more explicit where notable architectural assumptions are made.

---

> > > ### Comment · Reviewer_rSer · 2025-11-26
> > > **Update**
> > >
> > > Thanks for the changes. All my concerns are sorted. I'm recommend for acceptance.

---

### Author Response · Authors · 2025-11-21
**List of Changes**

Dear Reviewers,

Thank you for your careful review of our paper. We provide below a list of all changes made in this current revision, which will also be highlighted in red in the PDF.

# List of Changes

- expand exposition around transformer architecture in section 2.2
- Add in the ReLU / GELU approximation behavior in section 4.6
- fixed typos
- added citation to RASP/Tracr in the introduction
- added remark about architectural assumptions into the introduction
- fixed an error in section 5.7
- added comment about temperature scaling into section 9.1.1
- updated table 2 and created table 3

---

### Decision · Action_Editor_uvDz · 2025-12-22

**Recommendation:** Accept as is

**Audience:**

Yes

**Audience Explanation:**

Yes, the paper is potentially interesting to readers in several subfields covered by TMLR. Given the collection of building blocks and the practical focus, the paper may become somewhat of a practitioner's standard reference.

**Claims And Evidence:**

Yes

**Claims Explanation:**

The paper has compiled a list of practical techniques and building blocks to implement algorithms with transformers, or transformer-circuitry. While most of the material has been reported before, these reports are scattered across the literature and often embedded in papers that discuss more than the actual implementation. Compiling such a list is thus a sound and novel contribution, and it is well executed in the current manuscript (which adds quite a bit of discussion and practical considerations beyond mere reporting of the building blocks).

Some initial criticism of reviewers could be resolved, and after reading the revised manuscript and author responses all reviewers agree that the paper's claims are supported by evidence and that the paper should be accepted. I agree with that verdict.